# Spatial Augmented Reality for Expanding the Reach of Individuals with Tremor beyond Their Physical Limits

**DOI:** 10.3390/s24165405

**Published:** 2024-08-21

**Authors:** Kai Wang, Mengjing Wu, Zhi Sun, Qun Huang

**Affiliations:** School of Art and Design, Wuhan University of Technology, Wuhan 430070, China; kai_w@whut.edu.cn (K.W.);

**Keywords:** spatial augmented reality, tremor, eye-tracking, Internet of Things, assistive technology

## Abstract

Tremor is a prevalent neurological disorder characterized by involuntary shaking or trembling of body parts. This condition impairs fine motor skills and hand coordination to varying degrees and can even affect overall body mobility. As a result, tremors severely disrupt the daily lives and work of those affected, significantly limiting their physical activity space. This study developed an innovative spatial augmented reality (SAR) system aimed at assisting individuals with tremor disorders to overcome their physical limitations and expand their range of activities. The system integrates eye-tracking and Internet of Things (IoT) technologies, enabling users to smoothly control objects in the real world through eye movements. It uses a virtual stabilization algorithm for stable interaction with objects in the real environment. The study comprehensively evaluated the system’s performance through three experiments: (1) assessing the effectiveness of the virtual stabilization algorithm in enhancing the system’s ability to assist individuals with tremors in stable and efficient interaction with remote objects, (2) evaluating the system’s fluidity and stability in performing complex interactive tasks, and (3) investigating the precision and efficiency of the system in remote interactions within complex physical environments. The results demonstrated that the system significantly improves the stability and efficiency of interactions between individuals with tremor and remote objects, reduces operational errors, and enhances the accuracy and communication efficiency of interactions.

## 1. Introduction

Tremor, a prevalent neurological disorder, is characterized by involuntary oscillations or tremors of various body parts, affecting millions of individuals worldwide [1]. It can be broadly categorized into two primary types on the basis of their occurrence: resting tremors and action tremors [2]. Resting tremors, commonly seen in Parkinson’s and Wilson’s diseases, occur when the affected body parts are at rest. Parkinson’s disease, for instance, is marked by resting tremors that impact not only the limbs but also the head, hindering fine motor skills and coordinated movements such as stable walking or precise actions [2,3]. Wilson’s disease often leads to uncontrollable “wing-beating” resting tremors [4]. In contact, action tremors are primarily present during conscious movements or while maintaining certain postures, with essential tremor (ET) being a prime example. ET mainly affects the rhythmic movements of the upper limbs and hand, typically oscillating between 4 to 12 Hz, and may progress to involve the voice, head, or other body parts [2,5,6]. Regardless of their type, tremors significantly disrupt patients’ daily lives. The involuntary shaking can hinder walking or manipulating objects, and hand tremors can interfere with using everyday items such as remote controls or switches. These physical limitations often lead to reduced quality of life and increased dependence on others, sparking a growing demand for innovative solutions to help these individuals maintain independence and enhance their quality of life.

Despite the proposition of various medical treatments such as medications, deep brain stimulation (DBS), and physical therapy to alleviate the symptoms of tremor, they come with their own sets of challenges. Medications may cause side effects such as dizziness, fatigue, and nausea [7,8]. DBS, while effective, carries surgical risks and can impact speech and balance [9]. Physical therapy, although beneficial, might not always suffice to fully counteract the symptoms of tremor. Given these challenges, the development of assistive technologies not only stands as a vital complement to traditional treatments but also represents a crucial frontier in empowering patients to lead more autonomous and fulfilling lives.

Currently, most research on assistive technologies for tremors has primarily focused on developing anti-tremor devices. These devices, whether integrated into tools or worn on the limbs, effectively counteract the impact of hand tremors on physical operations, such as using a spoon, and have demonstrated effectiveness [10,11]. This line of research prioritizes improving the accuracy of detecting tremor signals and developing mechanisms that can produce counter-vibrations to maximize the cancellation of tremors. However, these approaches have limitations, as they require the use of customized tools or the wearing of specialized anti-tremor devices, which not only limit the universality of the technology but may also impose additional physical burdens and restrict activities. Moreover, tremors not only directly affect hand interactions but also impact the lower limbs, making it difficult for patients to move and indirectly hindering their effective interaction with their surroundings and distant objects. Traditional technologies typically address only the interactions within the hand’s reach and do not extend to areas beyond the body’s reach. Therefore, there is a need to develop a new technological approach to overcome these limitations, to enhance the universality of the technology, and to reduce the physical burden and interaction restrictions on the tremor patients.

Spatial augmented reality (SAR) is a branch of augmented reality (AR) that seamlessly integrates digital information into the physical environment without requiring any display devices to be worn by the user. SAR, through projectors, cameras, and other sensor technologies, superimposes virtual content directly onto physical objects or spaces, effectively “enhancing” the real world [12]. Its main advantage lies in supporting the users’ observations and direct interactions through natural viewpoints and achieving perfect integration with the environment, providing an immersive experience and immediate interaction feedback. Moreover, SAR’s projection technology has a broad coverage range, allowing users to effectively interact with the physical environment even from a distance. In light of these features, this study introduces a SAR system based on eye-tracking technology, designed specifically for individuals with tremors. This system allows them to control a virtual hand or pointer in the physical space environment using their eye movements, combined with Internet of Things (IoT) technology, to interact with objects and home appliances at a distance. In the design of the SAR system’s interactive controls, we fully considered the individual differences among tremor patients. Since the tremors of individuals vary greatly in terms of the symptoms, severity, and affected areas, it is difficult to find a universally applicable method to address all limb tremors. However, head tremors usually fall within a predictable range, and even patients with Parkinson’s tremors can counteract them with suitable stabilization techniques. Therefore, we used the method of controlling virtual objects with eye movements, combined with the development of virtual stabilization algorithms, to achieve “interact where you look”. This ensured that individuals with tremor can interact directly and steadily with their environment, extending beyond their physical limits and expanding their range of activity. Building on this foundation, the following innovations are introduced:A novel SAR system design: This study designed a new type of SAR system specifically for individuals with tremor. This system supports the users in interacting with objects in the living environment through eye-controlled virtual hands or pointers, without the need for wearing devices. This design significantly enhances their interaction capabilities with the surrounding environment and physical objects, reduces the limitations of body tremors on interactions, and expands the range of interaction.Virtual stabilization algorithm: This study also designed a virtual stabilization algorithm specifically for eye movement control by individuals with tremor. This algorithm ensures stable control of the spatial movement of the virtual hand or pointer, even in cases of involuntary head tremors.Experimental verification and application expansion: Through experimental verification, this study demonstrated the effectiveness of the virtual stabilization algorithm and the system in assisting individuals with tremor to perform complex interactive tasks and interact with complex environments. Additionally, through a series of examples, this study showcased the system’s expandability and diverse applications in assisting tremor patients to overcome physical limitations and promote long-distance interactions.

Furthermore, our system has the potential to assist those with other movement disorders, such as amyotrophic lateral sclerosis (ALS) or hand impairments such as claw hand, broadening the scope of its application and impact.

## 2. Related Studies

Given the limited efficacy of existing medical interventions in managing tremors, extensive research has been conducted to explore supportive methods for patients, seeking to bridge the gap between lifestyle interventions and traditional treatments. Functional electrical stimulation (FES) is a well-established approach, which utilizes electrical stimulation of the motor nerves to induce muscle contractions and counteract tremors. This technique, introduced by Prochazka et al. in 1989, has been widely adopted in clinical practice [13]. Initial clinical evaluations revealed an effectiveness rate of 73% for ET, 62% for Parkinson’s disease tremors, and 38% for cerebellar tremors [14,15]. Enhancements of the efficacy of FES for tremor suppression have been significant through the introduction of adaptive and sophisticated algorithms; however, this approach may lead to adverse effects such as trauma and physical discomfort [16].

Another effective, low-risk method for tremor suppression is the use of wearable orthotics, categorized into passive, semi-active, and active types [10,11]. Passive orthotics utilize spring damping or other mechanical structures to absorb the tremors’ energy [17,18]. Semi-active orthotics use sensors to monitor the tremors’ intensity and adjust the system’s impedance to mitigate it. The viscosity of magnetorheological fluids changes when exposed to a magnetic field, allowing the damping force to be controlled by adjusting the magnetic field’s strength [19,20]. Although passive and semi-active mechanisms reduce tremor movements by absorbing energy, they may also restrict voluntary movement due to the resistance imposed and struggle to adapt to the dynamic nature of tremors [11]. In contrast, active orthotics proactively counteract tremor movements by providing motions opposite to those of the tremor. As part of the Dynamically Responsive Interventions for Tremor Suppression (DRIFTS) project in 2005, the Wearable Orthosis for Tremor Assessment and Suppression (WOTAS) exoskeleton was developed. It includes sensors to measure rotational movements of the joints, a DC motor that converts electrical into mechanical energy, and a controller, all working in concert to suppress tremors in the wrist and elbow joints. The WOTAS achieved an effectiveness of 40% in suppressing tremors [21]. Subsequent developments in active orthotics have maintained similar design principles and mechatronic integration but have enhanced actuator designs to reduce weight and improve capabilities for tremor suppression [22,23]. However, these active orthotics are typically bulkier than their passive or semi-active counterparts, averaging around 561 ± 467 g. The number of degrees of freedom in these orthotics usually correlates with their weight, with most devices supporting only one or two degrees of freedom [11]. Biswas et al. [24] developed a wristband device that integrates accelerometer sensors with a pretrained machine learning model to detect hand movements and assess the tremors’ severity. This device dynamically adjusts the vibrating motors on the basis of accelerometer data to provide real-time haptic feedback that neutralizes or diminishes tremors. Nevertheless, this study noted that the wristband is primarily designed to alleviate hand tremors, and its effectiveness may be limited for Parkinson’s disease patients experiencing full-body tremors. While these technologies are adept at suppressing localized tremors, they commonly face limitations such as excessive weight and restricted mobility, which could compromise long-term wearability and practicality. Additionally, their capacity to manage whole-body tremors is somewhat restricted, particularly in complex neurological conditions such as Parkinson’s disease, thus potentially failing to offer comprehensive support.

Some studies have reduced the physical burden on tremor patients by incorporating anti-tremor modules into tools. For instance, Taşar et al. [25] designed a tremor-stabilizing spoon called FiMec, which helps patients with hand tremors eat more easily. It uses inertial measurement unit sensors to detect vibrations and controls two motors via a PID controller to absorb horizontal and vertical vibrations, thus stabilizing the spoon. Although FiMec has been shown to absorb between 84% and 99.409% of vibrations in laboratory tests, its validation in actual patients has not yet been conducted. Another commercially available tremor-stabilizing spoon, the Gyenno Spoon, was clinically tested by Ryden et al. to evaluate its effectiveness for patients with Parkinson’s disease tremors. Despite its intent to assist with eating, the results indicated that its effectiveness was limited, and in some tests, the amount of rice transferred using the device even decreased. This suggests that for patients with resting and low-amplitude postural tremors, the device may provide minimal assistance [26]. While such assistive tools are beneficial in specific domains such as eating and writing, their universality in other daily activities is limited.

Recent studies have made preliminary progress in assisting tremor patients with conventional devices. IBM [27] developed an assistive adapter that, once inserted between a mouse and a computer, provides digital motion smoothing, reduces unintended mouse clicks, and enhances double-click functionality, significantly improving the users’ experience. Additionally, Plaumann et al. [28] applied special filtering techniques to effectively mitigate tremors’ effects when using smartphones. Their development of motion sensors and optimized algorithms significantly enhanced the operational precision and response speed of tremor patients on smartphones. Wacharamanotham et al. [29] confirmed that, for patients with hand tremors, interacting with touchscreens through sliding gestures is more effective than tapping alone. However, despite these technological innovations enhancing the user’s experience for tremor patients, their market availability remains limited, failing to meet the needs of daily life.

In recent years, AR technology has shown tremendous potential in aiding tremor patients. Wang et al. [30] developed a low-cost rehabilitation training system specifically designed for Parkinson’s disease tremor patients, utilizing AR technology to provide professional rehabilitation guidance at home, which has proven more effective than video training in improving patients’ rehabilitation outcomes and experience. Innovations in AR increasingly utilize simultaneous localization and mapping (SLAM) technology to enhance users’ interaction through precise environmental mapping and device localization. For instance, Chen et al. [31] optimized SLAM in resource-limited settings by intelligently selecting keyframes and managing map constructions, significantly reducing the uncertainty of pose tracking to ensure stable AR interactions. Zhou et al. [32] developed a mobile AR system that dynamically updates 3D maps with digital twin technology and adaptive algorithms to maintain accuracy and responsiveness, even under fluctuating network conditions and varying movements by the user. Although these advancements represent significant progress, their deployment in indoor environments encounters challenges such as inconsistent network availability, computational constraints, and environmental variability, which can compromise the accuracy and reliability of AR systems. Wang et al. [33,34] demonstrated the application of AR in assisting tremor patients with typing, proving its feasibility in enhancing typing efficiency on conventional keyboards and reducing output errors. Additionally, Ueda et al. [35] introduced an “extended hand” system, allowing users to remotely control a virtual hand through a touch panel, opening new avenues for individuals with limited mobility to manipulate remote objects. However, this mode of hand interaction is not suitable for all tremor patients, particularly those with severe tremors or Parkinson’s disease tremors, who may face additional operational challenges and fatigue risks when using tablets. Future research should focus on exploring more natural and accessible remote interaction methods.

Eye-tracking technology in human–computer interactions primarily focuses on integrating data on eye movements into multimedia communications to create a seamless and intuitive user experience [36]. Increasingly regarded as a key real-time input method, it is particularly suited for patients with motor impairments such as amyotrophic lateral sclerosis (ALS) [37]. As technology advances, gaze-based human–computer interactions have expanded, giving rise to numerous innovative applications [38]. For example, Nehete et al. [39] developed an eye-tracking mouse that allows users to operate a computer via eye or nasal movements. Similarly, Missimer et al. [40] designed a system that controls the mouse cursor through the head’s position, with eyelid blinks emulating left and right mouse clicks. Additionally, the cheek switch technology, once used by Stephen Hawking, combines slight movements of the facial muscles with an infrared emitter to select letters for typing [41]. Although eye-tracking technology has proven its value in assisting ALS patients and other individuals with motor impairments in crucial communication interactions, its potential to support tremor patients remains to be further explored.

In summary, this study focused on utilizing SAR and eye-tracking technology to develop a more natural interaction method to assist tremor patients. It aimed to expand patients’ capabilities for interaction, enabling them to manipulate everyday objects and communicate effectively with minimal physical effort, despite limited mobility. Through this research, we aimed to create more adaptive, intuitive, and user-friendly assistance solutions to enhance the quality of life for tremor patients and overcome the limitations of traditional assistance methods.

## 3. Pilot System

To transcend the physical limitations of individuals with tremor and to expand their range of activity, we introduced a pilot SAR system specifically designed to assist individuals with tremor in interacting with objects and devices in their immediate environment. Recognizing that head tremors, irrespective of the type of tremor, typically remain within a relatively manageable range of instability, we developed an eye-controlled AR projection system. This system harnesses technology tracking eye movements to pinpoint the target of the user’s gaze, integrating it with IoT technology tailored for physical objects. Consequently, the users can seamlessly manipulate objects in the real world through simple eye movements, circumventing the physical strain and burden associated with conventional tremor assistance technologies.

### 3.1. The System’s Configuration

As illustrated in Figure 1, the hardware portion of the system comprised several crucial components: a computer, a projector (wheelchair compatible), an eye-tracking module (Tobii Eye Tracker 5), a control module (Arduino Nano, the IoT module of ESP-8266, an HX1838 infrared receiver, and an infrared emitter, etc.).

The system used the eye tracker for real-time capturing of data on eye movements, specifically targeting individuals affected by head tremors. The captured data underwent virtual stabilization processing, accurately identifying the stable gaze position of the eyes. Furthermore, the system has interactive interfaces for various tasks, such as the virtual hand and bubble. These features enable individuals suffering from tremors to interact with distant targets solely through their eye movements.

### 3.2. The Assistive User Interface

#### 3.2.1. The Virtual Hand

Using fingers for direct interaction is an intuitive and natural human behavior. However, individuals with tremors often struggle with manual dexterity due to involuntary bodily tremors. These tremors can significantly hinder their ability to engage with objects at a distance. In response, exploring the use of alternative limbs for interaction can help overcome these limitations, enabling stable and remote interactions.

To realize this objective, a system incorporating a projected virtual hand was meticulously designed to act as a surrogate for physical limbs, thereby aiding individuals with tremors in conducting remote interactions with objects. As depicted in Figure 2, the individual with trembling is seated in a wheelchair equipped with both a projector and an eye-tracking device. By adjusting the wheelchair’s orientation and deploying the eye tracker, he/she can effectuate control over a virtual hand through ocular movements, thereby enabling nuanced interaction with distant objects.

It is imperative to highlight that the eye tracker functions by recording the coordinate positions of the eye’s pupil, subsequently determining the virtual hand’s coordinate positions through spatial transformation techniques. This process facilitates the registration of the virtual hand within the physical space through a mapping mechanism between the virtual hand and the actual environmental space. Nevertheless, tremors in the head region could introduce interference in the signals captured by the eye tracker. This interference, once transmitted to the virtual hand, may be amplified, resulting in visual instability of the virtual hand. To mitigate the adverse effects of head tremors on the stability of the virtual hand, a virtual stabilization algorithm was incorporated; its specifics are elaborated in a later section. Through the implementation of this innovative technique, subjects can seamlessly control a stable virtual hand for purposes of interaction. When combined with the wheelchair configuration, this approach markedly extends the interaction radius accessible to individuals with tremors.

#### 3.2.2. Virtual Confirmation

To enhance the precision of daily interactive inputs in our system, we innovated a “bubble confirmation” mechanism. This feature is particularly useful in standard operations, such as button clicking, where the users traditionally confirm their inputs through prolonged pressing. Mirroring this in a virtual environment, our system introduces a bubble prompt that activates when the virtual hand or pointer hovers over a target. The process, illustrated in Figure 3, is straightforward: upon positioning the virtual hand or pointer over a target, a bubble encircles it. Inside this bubble, a blue dial gradually appears, filling the bubble in a sector-filling fashion over a set time period. For instance, when interacting with a television, this process spans approximately 3 s. This duration is meticulously calibrated to ensure not only the input’s accuracy but also the fluidity of the interaction. Should the virtual hand or pointer deviate from the target, the bubble reverts to its original state. This bubble mechanism is a key component in our system, reliably ensuring precise confirmation and gaze input for interactions with virtual objects.

#### 3.2.3. Assistive Interfaces

To facilitate easier operation of household appliances for individuals with tremors, some assistive interfaces were designed. This interface suite aimed to simplify the steps of interactions, making the operation process more intuitive and straightforward. Taking operation of a television as an illustrative example, an assistive interface, as demonstrated in Figure 3, was created. To ease the operation, this interface retains only the frequently used function modules, such as volume adjustment, switching channels, and the confirmation button. The assistive interface is projected around the television through a projector installed on a wheelchair. Individuals with tremors can interact with the assistive interface by controlling virtual hands and bubbles through their eye movements. All information on the interactions is transmitted in real-time to the television via microcontrollers and IoT modules, thereby controlling the television. This design can replace traditional remote control input methods, providing convenience for individuals with tremor who may find operating traditional controls challenging.

While such assistive interfaces represent a significant advancement, they still require careful customization. This customization must be based on the specific interaction patterns of eye movements, environmental contexts, and the functionalities of household appliances, with a particular focus on the needs of individuals with tremors. Such a tailored approach is crucial to enable them to effortlessly control various home appliances, thereby significantly enhancing their daily living experience and independence.

### 3.3. Virtual Stabilization

Our objective was to develop a virtual stabilization algorithm aimed at minimizing the impact of head tremors on the tracking of an eye tracker, leading to more stable eye-tracking. This algorithm is instrumental in differentiating between involuntary head tremors and intentional head movements. By effectively distinguishing these two types of motion, our system ensures smooth and stable control. This distinction is crucial, as it allows the system to respond accurately to the user’s intended movements, while simultaneously filtering out the disturbances caused by tremors.

Initially, the coordinates of the user’s eye movement are recorded at a specific moment as $\left(x_{0},y_{0}\right)$, along with the first three preceding coordinates: $\left(x_{1},y_{1}\right)$, $\left(x_{2},y_{2}\right)$, and $\left(x_{3},y_{3}\right)$. A set of weighted values, a, b, and c, respectively, is assigned to these coordinates to calculate the weighted average coordinates $\left(x^{\prime },y^{\prime }\right)$. Experimental testing determined that the optimal coefficients for a, b, and c, are 0.1, 0.8, and 0.1, respectively. The formula for this calculation is as follows:(1)$$x^{\prime }={a\;x}_{1}+{b\;x}_{2}+{c\;x}_{3}\;y^{\prime }={a\;y}_{1}+{b\;y}_{2}+{c\;y}_{3}$$

We then set a threshold distance value, denoted as d. The system compares the Euclidean distance between the initial coordinates $\left(x_{0},y_{0}\right)\;$ and the weighted average coordinates $\left(x^{\prime },y^{\prime }\right)$ with this threshold value. If the distance is greater than d, the system outputs the original coordinates $\left(x_{0},y_{0}\right)$. Conversely, if the distance is less than or equal to d, the system outputs the weighted average coordinates $\left(x^{\prime },y^{\prime }\right)$. The formulas for these conditions are:(2)$$IF\;\;\sqrt{{\left(x_{0}-x^{\prime }\right)}^{2}+{\left(y_{0}-y^{\prime }\right)}^{2}}\leq d,THEN\;\;x=x^{\prime },y=y^{\prime }\;IF\;\;\sqrt{{\left(x_{0}-x^{\prime }\right)}^{2}+{\left(y_{0}-y^{\prime }\right)}^{2}}>d,THEN\;x=x_{0},y=y_{0}\;$$

### 3.4. Simulation of Tremors

Considering the experimental system phase and ethical implications, it was not advisable to directly conduct experiments on patients with tremors at the initial stage. Additionally, to gain a more accurate understanding of how various characteristics of tremors match the system, utilizing a simulation method could more conveniently obtain objective usage data. Hence, we designed a tremor simulator enabling individuals without tremor to mimic those with trembling when testing the system. The foundational idea behind this simulation is that the effects produced by the shaking of a person’s head in front of a motion-tracking device are analogous to those of an eye-tracking device during trembling. As such, our design merely required the control of the eye-tracking device to tremble according to specific tremor frequencies and amplitudes.

The tremor simulator’s design comprised a wooden stand, a 3D-printed fixed head, an Arduino, and a jitter motor, collectively forming an eye tracker stand (Figure 4a). By swinging the fixed head of the stand up and down, we simulated the head tremors seen in patients with Parkinson’s disease. According to Xu’s experiment [42], the average amplitude of tremors in patients with Parkinson’s disease is 1–2 cm and the frequency is 5–6 Hertz. Therefore, imitating the characteristics of this type of tremor population, the swing amplitude of the motor-controlled eye tracker was controlled within the range of 2 cm and 4–8 Hertz. To ensure that the simulated data were consistent with the characteristics of tremors, we used an Opti-track device to track the jitter at the center point of the eye tracker and monitor its motion trajectory, as shown in Figure 4b.

## 4. Experiments and Results

This study was structured around three experiments, designed to comprehensively assess the practicality and auxiliary benefits of the pilot system. Firstly, Experiment 1 focused on evaluating a virtual stabilization algorithm within the SAR system. Specifically, we aimed to test the stability and efficiency of interactions with the target objects by individuals with trembling using the SAR system equipped with this virtual stabilization technology, alongside assessing the system’s adaptability to eye movement control by individuals with tremors. Subsequently, Experiment 2 delved into exploring the performance of operations when individuals with tremors used the system to execute relatively complex interactive tasks. This phase evaluated the system’s efficacy in assisting patients during tasks that demand a higher degree of operational complexity. Lastly, Experiment 3 aimed to investigate the precision of pointing and the effectiveness of distant interactions in complex physical environments by individuals with tremors using the system. We were particularly interested in determining whether the system enhanced the individuals’ ability to interact with distant targets and improved the efficiency of their communication within the AR environment. To conduct these experiments, we invited volunteers to participate in rigorous simulated tests. Both subjective experiences and objective data were systematically collected and analyzed for a comprehensive evaluation.

### 4.1. Evaluating the Virtual Stabilization of the SAR System

This study aimed to validate the virtual stabilization algorithm in the system assisting individuals with trembling to interact stably and efficiently with distant objects through eye movement control, such as virtual hands, pointers, and bubbles.

#### 4.1.1. Experimental Setup

As illustrated in Figure 5, participants were tasked with controlling a virtual pointer through eye movements, aiming to rapidly and steadily align the pointer with randomly appearing red balls in physical space. The experimental procedure encompassed several key steps. At the beginning of each trial, a red dot was randomly generated on the interface. The participants were required to maintain the eye-controlled bubble pointer within the collision range of the red dot for 3 s, after which, the red dot would disappear and reappear at a new location. Each participant repeated this task five times. Ten participants aged between 20 and 80 years were recruited for the study. Prior to the experiments, the simulator’s frequency and amplitude of vibration were precisely calibrated to mimic the characteristics of real head tremors.

Two experimental conditions were set up, specifically one with the system equipped with the virtual stabilization algorithm and the other without it, to compare the effects. The system automatically recorded the coordinates of the pointer and the target’s position, as well as the task completion time during each trial. Upon completion of the experiments, subjective assessments were collected from the participants to measure their personal experiences and perceptions of the experiment. These assessments were rated on a seven-point Likert scale, ranging from −3 (strongly disagree) to +3 (strongly agree), allowing the participants to express their level of agreement or disagreement with each statement. The assessment questions included the following.

Q1: The movement of the virtual pointer was entirely controlled by my eye movements.

Q2: I experienced stability while manipulating the virtual pointer through my eye movements.

Q3: I was capable of accurately targeting objects with the eye-controlled virtual pointer.

#### 4.1.2. Results

This study assessed the effectiveness of SAR systems integrated with virtual stabilization algorithms in improving both the efficiency and stability of users’ interactions. By analyzing objective data autonomously captured by the experimental setup, we compared the time taken by participants to complete the task of touching a distant random target and the average jitter distance of the virtual pointer during the process of targeting under two conditions: with and without the activation of the virtual stabilization algorithm in the SAR system.

Evaluation of the efficiency of virtual stabilization: A key finding of our research was a significant improvement in efficiency, measured as the reduction in the time spent on tasks, when the virtual stabilization system was used. The paired sample *t*-test demonstrated a statistically significant decrease in time (*p* = 0.000247) with the implementation of the system. Specifically, as illustrated in Figure 6a, the mean time spent without the system was 94.31 s (SD = 29.21), which was significantly reduced to 36.72 s (SD = 8.43) with the system. The effect size, calculated as Cohen’s d, was a substantial 2.68, highlighting the system’s effectiveness in minimizing the task completion time.

Assessment of the stability of virtual stabilization: In terms of stability, measured as the average jitter distance of the virtual pointer, we obtained this measurement by randomly sampling the spatial coordinates of the virtual pointer during the process of targeting the red ball. The Wilcoxon signed-rank test revealed a statistically significant reduction when the virtual stabilization system was active (*p* = 0.0039) (Figure 6b). The results indicated that the mean distance covered was notably less when the system was utilizing the virtual stabilization algorithm compared with when it was not in use, further confirming the system’s role in enhancing operational precision and stability.

We analyzed the participants’ feedback on the three key questions, Q1, Q2, and Q3, to further evaluate the actual effectiveness of the virtual stabilization algorithm in the SAR system. Specifically, we examined how the algorithm impacted the users’ experience of individuals with tremor when they used eye control to interact with distant targets via a virtual pointer. Q1 focused on assessing the control strength of the virtual pointer, Q2 evaluated the stability of control, and Q3 examined the accuracy of control.

For Question 1 (Q1), which assessed the extent to which participants felt the movement of the virtual pointer was controlled by their eyes, the mean difference in the responses was 3.3 (95% CI: 0.09 to 6.51), with a significant *p*-value of 0.000043 (Figure 7). This indicated a considerably higher sense of control under the system with virtual stabilization.

Similarly, for Question 2 (Q2), regarding the stability experienced in controlling the virtual pointer’s motion, the mean difference was 4.4 (95% CI: 1.17 to 7.63), with an even lower *p*-value of 0.0000045 (Figure 7). This result underscores a marked improvement in perceived stability with the stabilization feature.

Question 3 (Q3), which focused on the participants’ ability to accurately target objects using the eye-controlled virtual pointer, also showed a significant mean difference of 4.1 (95% CI: 0.82 to 7.38) and a *p*-value of 0.000009 (Figure 7). This finding suggests that the virtual stabilization system substantially enhanced the accuracy of eye-controlled targeting.

### 4.2. Evaluating the System-Assisted Effectiveness of Complex Interactive Tasks

This study aims to validate the effectiveness of the SAR system in assisting individuals with tremors in operating everyday household appliances. While controlling a television may seem a common and straightforward task in daily life, it indeed comprises a variety of complex interactive tasks, such as switching functions, confirming selections, and returning to previous menus. Patients with tremor disorders often face challenges in smoothly completing these tasks using traditional control devices, such as TV remotes, due to hand instability. Given the representativeness and ubiquity of controlling televisions, it served as an ideal test case for assessing the efficacy of the SAR system in everyday interactive tasks.

#### 4.2.1. Experimental Setup

The design of our experiment was as follows. We selected several common TV control tasks, such as volume adjustment, switching channels, and confirmation. These tasks were arranged in a random sequence to create 10 sets of TV control task sequences. We recruited 10 participants aged between 20 and 65 years to complete these tasks. Each participant’s primary task involved using the SAR system and the conventional remote control to complete a randomly selected set of TV control tasks. For clarity in subsequent analysis, specific terminologies were designated for the experimental conditions: interactions facilitated by the SAR system were termed “system-assisted”, while those involving the conventional remote control were labeled “manual interaction”. To simulate the manipulation with a trembling finger typical of tremor patients when using a traditional remote control, we used a hand tremor simulator, describes as “a dual channel electrical stimulation instrument for simulating trembling limbs” [42].

Upon completion of the tasks, the participants were asked to respond to a series of subjective questions based on their experience. Responses were rated on a 7-point scale, which ranged from −3 (very poorly matched) to 3 (perfectly matched). Through these responses, we aimed to assess the potential effects of the SAR system in enhancing the operational accuracy and the users’ experience.

Q4. In this condition, switching channels and volume adjustment could involve a few mis-operations.

Q5: In this condition, switching channels and volume adjustment could be easily performed.

Q6: In this condition, switching channels and volume adjustment were performed stably.

#### 4.2.2. Results

In this part, we focused on evaluating the effectiveness of the SAR system in assisting individuals with tremors with complex interaction tasks. Specifically, Q4 assessed the frequency of misoperations with the assistance of the SAR system, Q5 examined the ease of interaction with the SAR system, and Q6 evaluated the stability of interactions performed with the assistance of the SAR system. Our statistical analysis utilized a paired *t*-test to compare the “system-assisted” and “manual interaction” conditions. Our analysis revealed a statistically significant improvement in the number of misoperations during the tasks of switching channels and volume adjustment when the participants used the SAR system as opposed to a conventional remote control.

This was evident from the responses to Question 4 (Q4), which showed a mean difference of 3.6 (95% CI: 0.19 to 7.01, *p* = 0.000035) (Figure 8). This underscores the potential of the SAR system to reduce misoperations in control tasks that are typically challenging for individuals with tremors.

However, when evaluating the ease of performing the same tasks, as assessed in Question 5 (Q5), the statistical analysis did not demonstrate a significant difference between the two conditions (mean difference: −0.2, 95% CI: −2.98 to 2.58, *p* = 0.619) (Figure 8). This suggests that participants did not perceive a change in the difficulty level when using the SAR system compared with the manual method.

In regard to the stability of operation, as queried in Question 6 (Q6), while the mean difference in the responses was 1.3, suggesting an improvement under “system-assisted” conditions, the results did not achieve statistical significance (95% CI: −3.23 to 5.83, *p* = 0.070) (Figure 8). This outcome implies a potential trend towards increased stability with the assistance of the SAR system; however, the data do not allow for a definitive conclusion, indicating the need for further research with a larger sample size or refined experimental conditions.

### 4.3. Evaluating the System-Assisted Effectiveness in Distant Interactive Environments

This experiment aimed to explore and validate the potential of SAR technology in assisting individuals with tremors in remote communication and interaction. For individuals affected by tremors, particularly those impacting hand stability, accurately indicating and interacting with distant objects poses a significant challenge. Limited by their physical mobility, these patients often rely on the assistance of and communication with others for interacting with objects that are out of reach, such as needing help to retrieve items located at a distance. The complexity of real-life environments, where objects may be obscured, closely packed, or small, further complicates this task. The experiment involved a pointing and guessing game, created to test the effectiveness of the SAR system in complex situations. The main objectives were to determine if the SAR system enhanced their ability to accurately and efficiently point at objects, and to assess how the system aided communication with others.

#### 4.3.1. Experimental Setup

In our experimental setup, we carefully selected 10 objects of various sizes commonly found in households, including items such as cardboard boxes, bottled water, apples, and wall paintings (Figure 9). Each object was assigned a unique numerical identifier to facilitate the experiment’s various stages. Spatially, five objects were placed on an upper tier and four on a lower tier, and the wall painting was hung on an adjacent wall. To increase the complexity of the task and more closely simulate a real-life environment, smaller objects were positioned in front of the larger ones on both tiers, creating a partial visual obstruction. All objects were strategically positioned 5 m away from the participants, covering a horizontal range of 0 to 0.5 m and a vertical range of 0 to 1 m.

The task involved a sequence of numerically coded objects presented in a randomized order. As illustrated in Figure 9, the experiment was conducted in 10 rounds, each involving two participants. One participant, using a hand tremor simulator or using the AR system, was to point to the distant objects. The other participant’s role was to guess and note the numerical codes of the objects indicated. Upon the conclusion of each round, we compared the recorded numerical sequences of both participants, focusing on the discrepancies and the total time taken for each pointing task. Additionally, participants provided feedback on their experience, rating it on a 7-point scale at the experiment’s conclusion. The scale ranged from −3 (very poorly matched) to 3 (perfectly matched), offering insights into their subjective experiences. The following questions guided their feedback:

Q7: In this case, I think I can more easily point to distant objects.

Q8: In this situation, I feel that I can stably point to the object.

Q9: In this case, I can easily understand the object being referred to.

Q10: In this situation, I feel that communication is easy.

#### 4.3.2. Results

This section focuses on evaluating the effectiveness of the SAR system in assisting individuals with tremors within complex distant interaction environments. We assessed the system’s performance by analyzing the objective data from remote pointing tasks, specifically evaluating the accuracy and efficiency of distant pointing. Since the participants were unable to successfully complete the tasks without the system, our analysis was based solely on the data collected under the system’s assistance.

Evaluation of the accuracy of distant pointing: As illustrated in Figure 10, the analysis of the data revealed that under the support of the system, the participants were able to achieve a median accuracy rate of 90%, with the first quartile at 80% and the third quartile reaching 100%. This high level of accuracy demonstrates the system’s effectiveness in aiding participants to overcome challenges posed by obscured, closely packed, or small objects.

Evaluation of the efficiency of distant pointing: The average time taken to complete the task showed a median of 3.65 s, with the bulk of the experiments falling between 3.0 s (first quartile) and 5.225 s (third quartile). This indicates a relatively quick response time in completing the tasks, highlighting the system’s role in facilitating efficient interactions.

Notably, our analysis did not find a significant direct correlation between the average time spent on tasks and the level of accuracy achieved. This indicates that the current level of accuracy was not attained at the expense of prolonged duration of the interaction. Overall, the experimental results demonstrated the system’s effectiveness in significantly enhancing remote communication for individuals with tremors. 

We also analyzed the participants’ feedback on key questions Q7, Q8, Q9, and Q10 to further demonstrate the practical effects of the SAR system in facilitating remote interactions for individuals with tremors. Specifically, Q7 assessed the ease of indicating distant objects, Q8 evaluated the stability of these indications, and Q9 provided additional verification of the effectiveness of remote interactive indications based on the understanding of the indications by peers. Q10 measured the ease of remote communication. All these evaluations were conducted with and without the assistance of the SAR system.

Question 7 (Q7) addressed the ease of performing remote pointing tasks. Participants noted that pointing at distant targets was easier when using an SAR-assisted system. Statistical analysis revealed a significant improvement in remote pointing ability with the system’s assistance, with a mean difference of 4.4 (95% CI: 1.35 to 7.45, *p* = 0.00000278) (Figure 11). This underscores the enhanced precision in locating distant objects by individuals with tremors with the aid of the system.

Question 8 (Q8) focused on the stability of participants while pointing at objects. The statistical outcomes indicated the significant aid provided by the system for individuals with tremors in performing stable pointing actions (mean difference of 3.2, 95% CI: −1.89 to 8.29, *p* = 0.0015) (Figure 11). This demonstrated that the virtual hand and pointer interface offered by the system can be controlled steadily by individuals with tremors.

Question 9 (Q9) explored the clarity of the pointing actions. The participants reported that it was easier to understand the indications given by individuals with tremors in a system-based environment (mean difference of 3.2, 95% CI: −1.42 to 7.82, *p* = 0.00079) (Figure 11). This highlights the effectiveness of using a virtual hand for remote interaction and pointing in the system’s setting.

Question 10 (Q10) examined the comparative ease of remote communication using the SAR system versus traditional methods. The responses indicated that the convenience of communication significantly improved with the use of SAR technology (mean difference of 4.0, 95% CI: −0.13 to 8.13, *p* = 0.0000685) (Figure 11), suggesting that the system facilitates clearer communication in interactive tasks. These results collectively emphasize the important role of SAR systems in enhancing the precision and convenience of interactions in complex environments.

### 4.4. Discussion

The results of the first experiment demonstrated that the SAR system using the virtual stabilization algorithm significantly reduced the time required for participants to complete tasks of pointing at remote target and markedly improved efficiency. Additionally, the average jitter distance of the virtual pointer during tasks was significantly reduced, further confirming the virtual stabilization algorithm’s effectiveness in enhancing the operational precision of the eye-controlled virtual hand and pointer. These findings underscore the critical role of the virtual stabilization algorithm in improving the stability and efficiency of interactions between individuals with tremors and distant objects through eye movement-based control.

In the second experiment, operations under the system-assisted condition significantly reduced errors compared with traditional manual interactions (Q4), highlighting the importance of the SAR system in enhancing operational precision, particularly when tremor patients use conventional control devices such as TV remotes. However, there was no significant difference in the ease of control (Q5) between the two conditions, suggesting that the system interface may not yet be fully optimized for eye movement-based interaction. Additionally, the results for operational stability (Q6) did not reach statistical significance, possibly due to insufficient sample size or the precision of the measurement tools.

The third experiment explored the performance of the SAR system in remote pointing tasks within complex environments. Supported by the system, the participants achieved a median accuracy rate of 90%, and the median task completion time was only 3.65 s, demonstrating the system’s ability to maintain high operational precision while ensuring a rapid response. The survey results further indicated that the system significantly enhanced the participants’ ability to easily point to distant targets (Q7), improved the stability of pointing (Q8), clearly conveyed pointing intentions to their peers (Q9), and substantially increased the efficiency of communication (Q10). These findings highlight the potential value of the SAR system in enabling precise and efficient interactions for tremor patients.

Despite these promising results, certain metrics, such as the ease of use of the SAR system in complex interaction tasks, did not show significant improvement. This suggests that future research should focus on developing assistive interface designs that are better suited to the characteristics of eye movement-based interactions to enhance the users’ experience. Additionally, a significant limitation of this study was the absence of other tremor assistance technologies as comparative baselines. This omission prevented a comprehensive evaluation of the SAR system’s relative advantages across various performance parameters. Therefore, future research should incorporate comparisons with other studies (such as [24,25]), including multiple baselines, to provide a more thorough assessment.

Although the complexity and diversity of head tremors are less than those of hand tremors, the virtual stabilization algorithm in this study effectively managed the frequencies and amplitudes of head tremors in the most common tremor patients. However, this design may not be sufficient for more severe types of tremors. Future systems should incorporate machine learning-based tremor measurement algorithms to assess mild, moderate, and severe tremors, thereby optimizing the selection and parameters of the virtual stabilization algorithm.

## 5. Application

This system demonstrated significant potential in aiding individuals with tremors, helping them overcome physical limitations and facilitating remote interactions. In the following sections, we illustrate, through a series of examples. how individuals with tremors can use this system for controlling home appliances and conducting daily communication within a household setting.

### 5.1. Light Control

Operating light switches is a commonplace interaction within households. However, for individuals affected by tremors, even the seemingly simple task of turning lights on and off can become challenging due to their physical constraints. Our system seamlessly integrates into the household’s lighting infrastructure, enabling users to control lights through their gaze instead of physical movements.

In terms of the system’s design details, the initial setup involved connecting the light switch to a relay. This relay was controlled by an Arduino mainboard equipped with a Wi-Fi module (ESP8266), facilitating intelligent control over the lighting. During the user’s interaction, if the user focuses their gaze on the virtual switch for 3 s, the system accurately recognizes and logs this interaction, and immediately sends a control signal to the Arduino mainboard. Upon receiving this signal, the mainboard promptly responds by converting it into an electrical operation, accurately activating or deactivating the light switch. This provides users with an indirect but precise method to control the lights via the virtual switch (Figure 12).

### 5.2. TV Control

Using television remotes can be a daunting task for individuals with tremor due to the precision and steadiness required. Our innovative system seamlessly integrates with televisions, allowing individuals with tremors to effortlessly control TVs without the need for physical remotes, simply by using their gaze.

During the initial setup, an Arduino motherboard was connected to an infrared transmitter, calibrated to match the signal of traditional TV remotes, enabling intelligent and responsive control through our system. As shown in Figure 13, the users efficiently interacted with the system by focusing their gaze on projected virtual buttons for 3 s, activating commands such as “confirm”, “back”, “channel”, “volume”, etc. This intuitive and user-centric design provides individuals with tremors with a straightforward and accessible way to accurately and effortlessly operate the TV through virtual controls.

### 5.3. Accessible Communication

Individuals with tremors often face challenges in movement and reaching items at height due to involuntary shaking of the body and limited mobility, such as wheelchair use, significantly impacting their daily communication. Discussing items located afar or at elevated positions becomes particularly challenging without assistance. For instance, discussing a painting hung high on a wall or instructing someone to retrieve an item from a distant location can be difficult for those with tremors.

However, these communication barriers were effectively eliminated with the use of our system. As illustrated in Figure 14, an individual with tremors could control a stable virtual hand through eye movements to communicate about a painting located at a height. As shown in Figure 9, they can also direct others to retrieve items by controlling the virtual hand. Additionally, when our system was integrated with wheelchair configurations, it enabled individuals with tremors to easily navigate and reach various corners within the home, facilitating barrier-free communication. This spatial prompting method simplifies the communication process and enhances clarity and efficiency.

The application of the system extends beyond the aforementioned scenarios to a wider range of living situations, including adjustment of air conditioning temperature, automated curtain control, and management of kitchen appliances. The system can be deeply customized and optimized based on the users’ specific needs, eliminating many barriers to interaction in daily life and expanding the range of interaction for individuals with tremors. By reducing the difficulty of daily tasks and decreasing reliance on external assistance, this system opens a new path for tremor patients towards a more independent, efficient, and comfortable lifestyle.

## 6. Conclusions

This study developed and evaluated an innovative SAR system, specifically designed to help patients with tremors overcome physical limitations and expand their range of activities. The system integrates eye-tracking and IoT technologies, enabling users to interact precisely and stably with remote objects through eye movements. Tremor simulation experiments confirmed that the virtual stabilization algorithm significantly enhanced the system’s operational stability and interaction efficiency. Additionally, the system demonstrated exceptional performance in assisting individuals with tremors with complex interactive tasks and adapting to complex environments. Furthermore, the research highlighted the potential of combining the SAR system with IoT technology, effectively overcoming the physical constraints of individuals with tremors and promoting diverse interactions with the surrounding environment. However, the current system’s design does not involve direct support during physical interaction for tremorous limbs, as it does not restrict the body’s movement. Future studies plan to clinically investigate individuals with tremors using the SAR system and quantitatively assess the system and its algorithms in terms of assistive performance, to further verify its practicality and effectiveness.

## Figures and Tables

**Figure 1 sensors-24-05405-f001:**
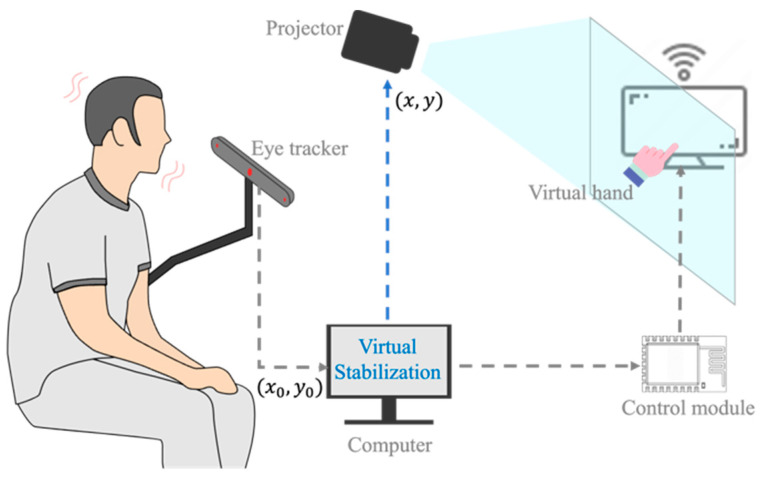
Configuration of the SAR system.

**Figure 2 sensors-24-05405-f002:**
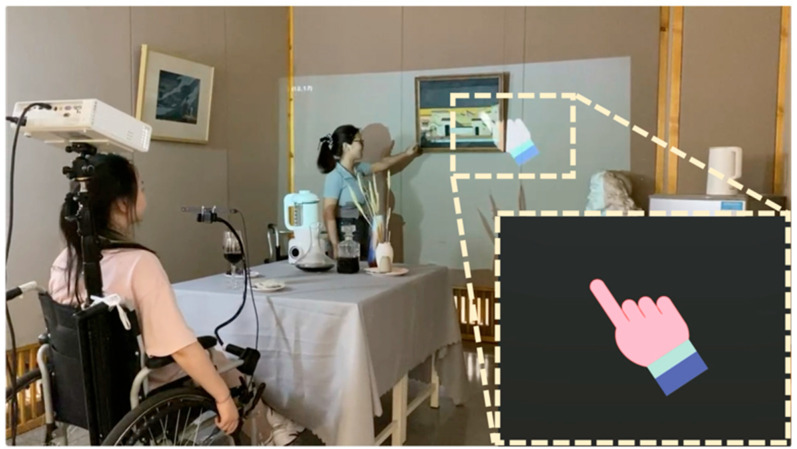
A virtual hand for interacting with distant real-world objects.

**Figure 3 sensors-24-05405-f003:**
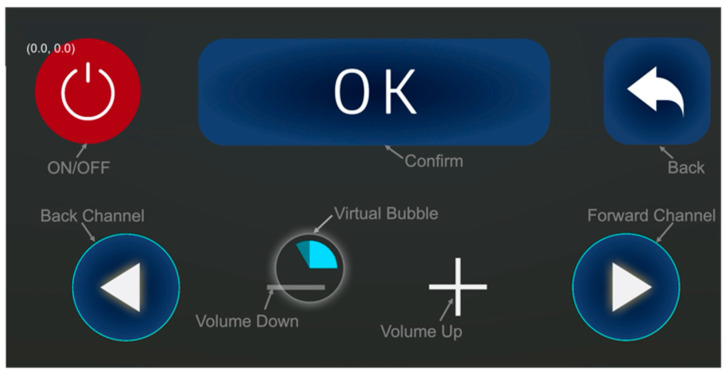
A virtual bubble and assistive TV interface.

**Figure 4 sensors-24-05405-f004:**
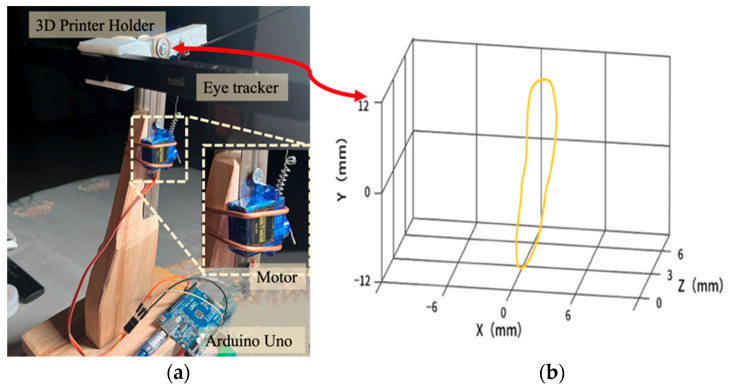
(**a**) The head tremor simulator. (**b**) Motion trajectories of the eye tracker.

**Figure 5 sensors-24-05405-f005:**
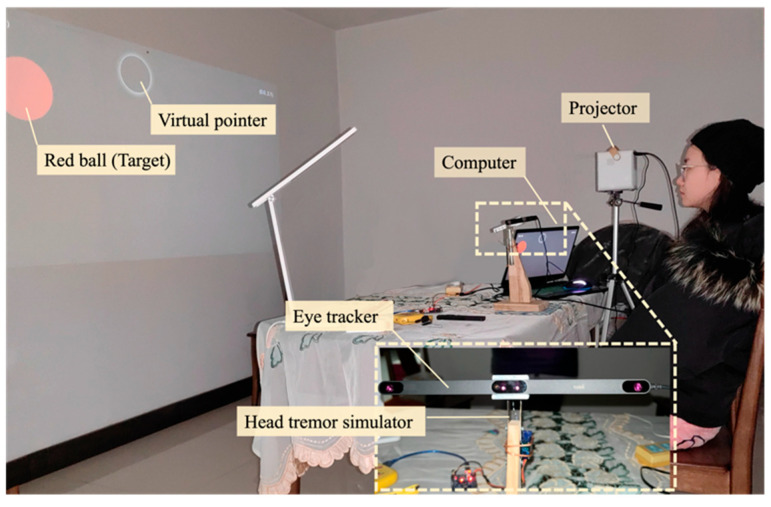
An experimental scenario for evaluating virtual stabilization.

**Figure 6 sensors-24-05405-f006:**
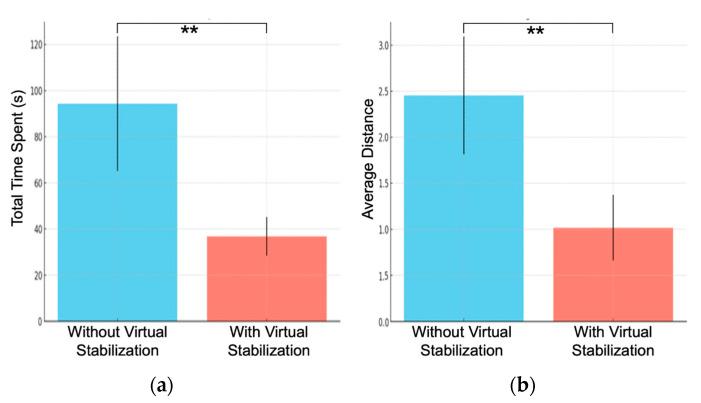
Comparison of (**a**) the mean time spent and (**b**) the average jitter distance during the virtual hand’s positioning of physical targets under the conditions of the system with and without virtual stabilization (** *p* < 0.01).

**Figure 7 sensors-24-05405-f007:**
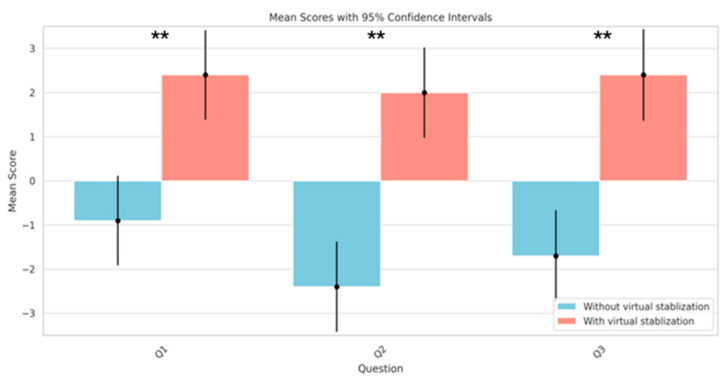
Comparison of the mean scores of Q1 to Q3 under the conditions of the system with and without virtual stabilization (** *p* < 0.01).

**Figure 8 sensors-24-05405-f008:**
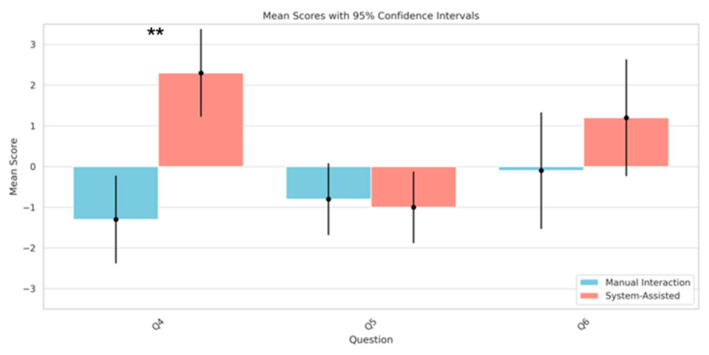
Comparison of the mean scores of Q4 to Q6 under the conditions of “system-assisted” and “manual interaction” tasks (** *p* < 0.01).

**Figure 9 sensors-24-05405-f009:**
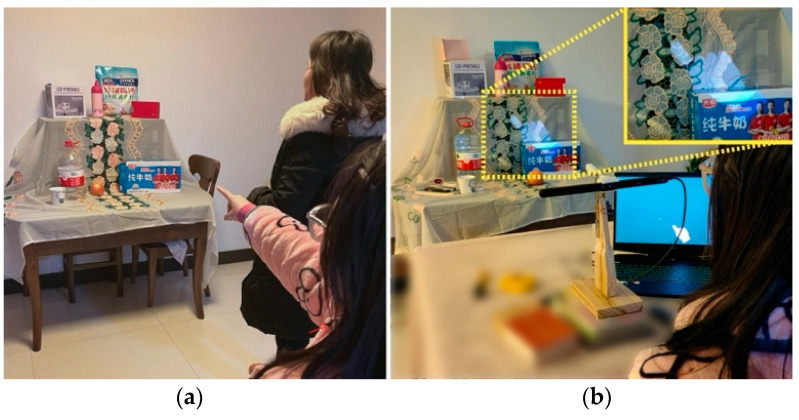
Evaluation of the system’s assistance in a distant interaction environment: (**a**) a participant directly indicating distant items; (**b**) a participant using the system’s assistance to indicate distant items.

**Figure 10 sensors-24-05405-f010:**
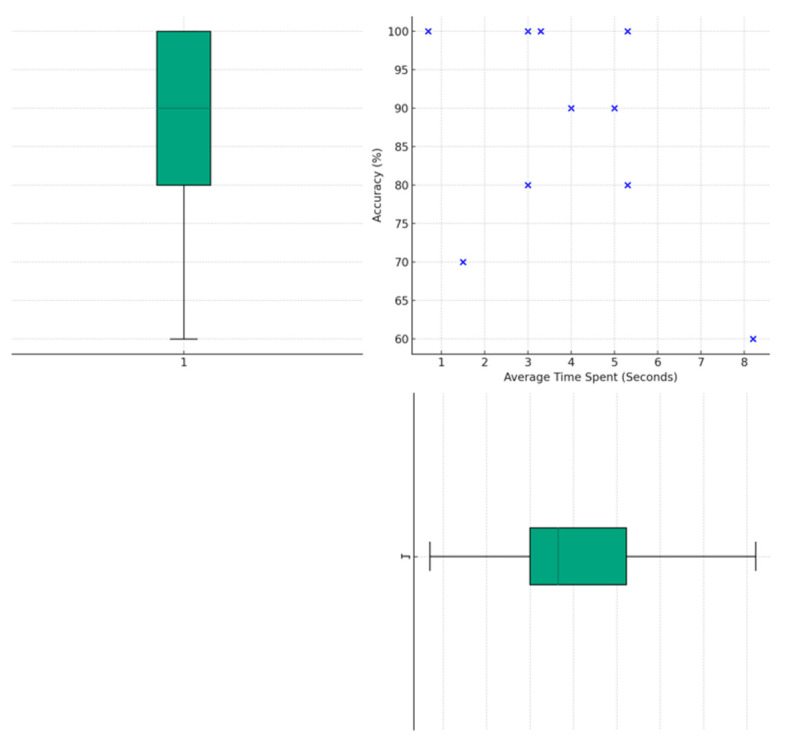
Assessing participants’ performance in distant interactive environments under the system’s support: accuracy and average time spent (The blue cross points represent individua participant results).

**Figure 11 sensors-24-05405-f011:**
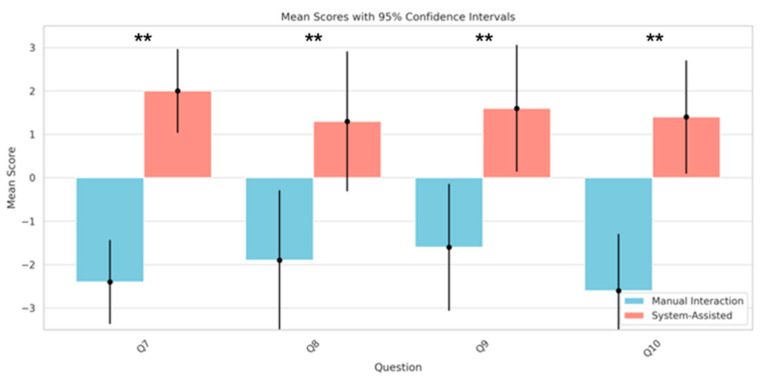
Comparison of the mean scores of Q7 to Q10 under the conditions of “system-assisted” tasks and “manual interaction” (** *p* < 0.01).

**Figure 12 sensors-24-05405-f012:**
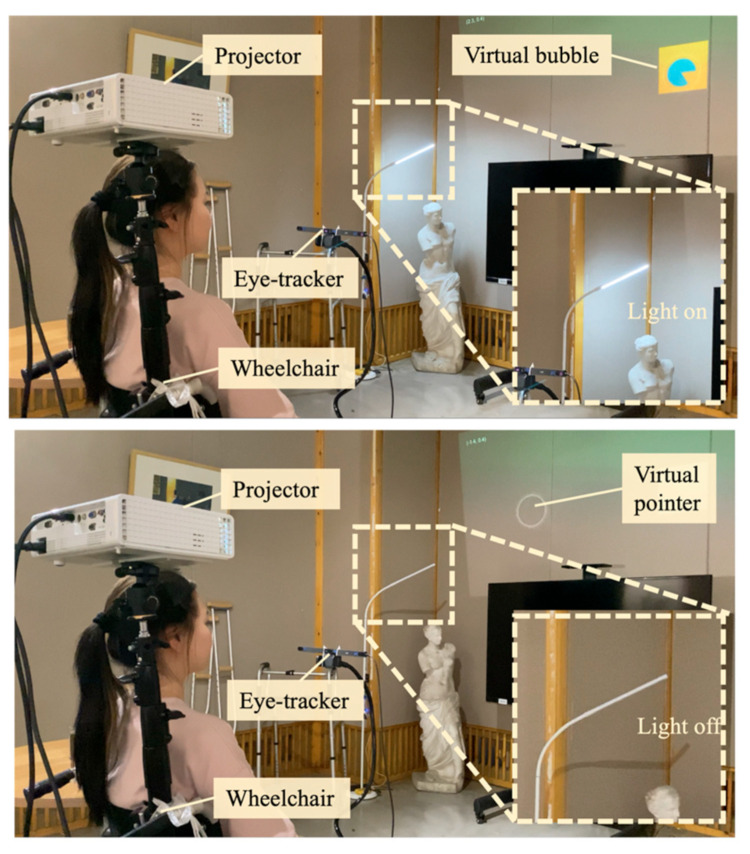
An individual with tremors using the SAR system for remote control of light switches.

**Figure 13 sensors-24-05405-f013:**
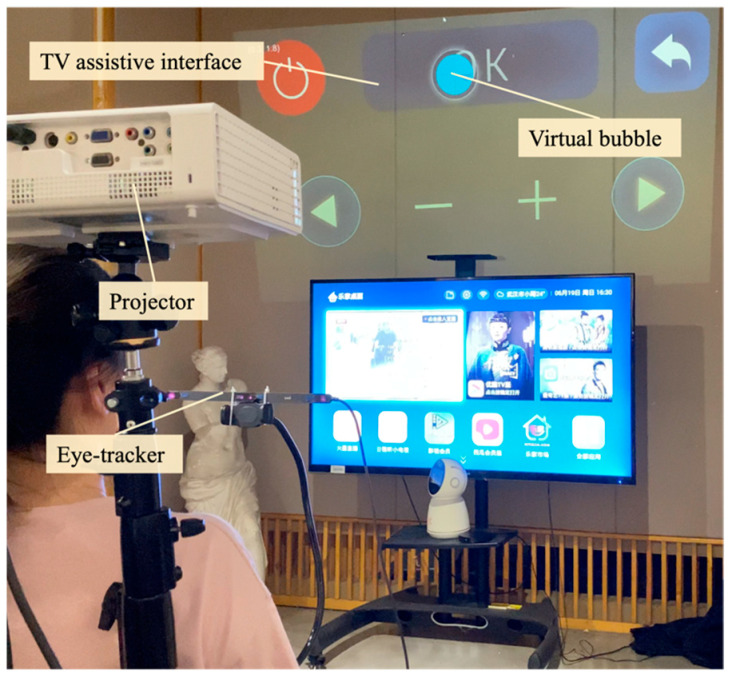
An individual with tremors utilizing the SAR system for remote operation of a television.

**Figure 14 sensors-24-05405-f014:**
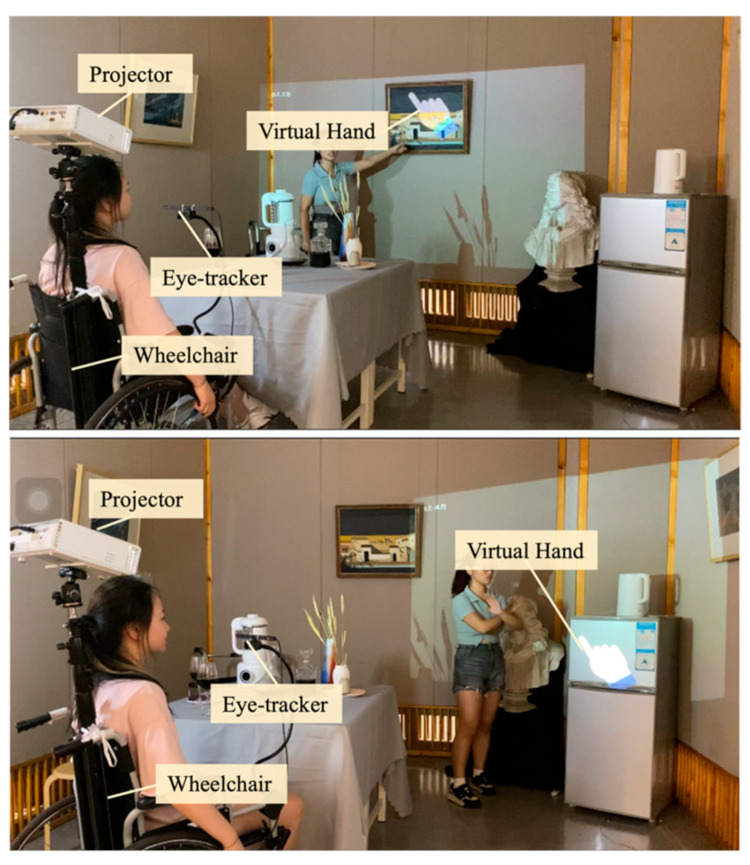
An individual with tremors engaging in daily communication with a companion using the SAR system, discussing distant objectives.

## Data Availability

The datasets generated during and analyzed during the current study are available in the “figshare” repository at https://figshare.com/s/875ec019dc0f6e7bae6e (accessed on 30 December 2023).

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
