# Peer review of "Spatial Augmented Reality for Expanding the Reach of Individuals with Tremor beyond Their Physical Limits"

_sensors, 2024, doi:10.3390/s24165405_

Round 1

Reviewer 1 Report

Comments and Suggestions for Authors

 This study developed a Spatial Augmented Reality (SAR) system to assist individuals with tremor disorders to overcome their physical limitations and expand their range of activities. The system integrates eye-tracking and Internet of Things (IoT) technologies, enabling users to smoothly control objects in the real world through eye movements. It employs a virtual stabilization algorithm for stable interaction with objects in the real environment.

Some questions and comments are listed below:

  1. While AR has been investigated in many areas, the defination of AR should be clarified in the introduction. Particularly, the definition of SAR should be clearly introduced for easy understanding.
  2. The authors only introduce some research topics related to AR in Section 2, rather than highlighting some differences.
  3. The contributions and motivation of this paper should be highlighted. For example, in this system, which part is challenging?
  4. The differences between this work and some state-of-the-art methods in AR are not compared. For example:

          AdaptSLAM: Edge-assisted adaptive SLAM with resource constraints via uncertainty minimization

         Digital Twin-based 3D Map Management for Edge-assisted Device Pose Tracking in Mobile AR

  1. In the existing experiments, the authors focus on the accuracy of eye tracking. However, the delay is also significant, especially for the design of the SAR system. How can the computing delay be tolerable for the considered application?
  2. The presentation should be improved, and some sentences are not clear. In addition, the citation in the paper cannot be displayed correctly. “[Error! Reference source not found.].” is shown.

Comments on the Quality of English Language
  1. The presentation should be improved, and some sentences are not clear. In addition, the citation in the paper cannot be displayed correctly. “[Error! Reference source not found.].” is shown.

Author Response

Dear Reviewer,

We sincerely appreciate the time and effort that the reviewers have invested in evaluating our manuscript. Your insightful comments and constructive suggestions have been invaluable in improving the quality and clarity of our work. We have carefully considered and addressed each point raised, and we believe that the revisions have significantly enhanced the manuscript. Thank you once again for your thorough review and valuable feedback.

Comments 1: [While AR has been investigated in many areas, the definition of AR should be clarified in the introduction. Particularly, the definition of SAR should be clearly introduced for easy understanding.]

Response 1: Thank you for your valuable feedback. We agree with this comment and have supplemented the definition of SAR in the introduction for clarity. The revised text is as follows:

[Spatial Augmented Reality (SAR) is a branch of Augmented Reality (AR) that seamlessly integrates digital information into the physical environment without requiring any display devices to be worn by the user. SAR, through projectors, cameras, and other sensor technologies, superimposes virtual content directly onto physical objects or spaces, effectively "enhancing" the real world [12]. Its main advantage lies in supporting user observations and direct interactions through natural viewpoints and achieving perfect integration with the environment, providing an immersive experience and immediate interaction feedback. …  -- This definition has been included on page [2], line [70] to [77].]

Comments 2: [The authors only introduce some research topics related to AR in Section 2, rather than highlighting some differences.]

Response 2: Thank you for highlighting this issue. We have revised the section to better emphasize the differences and advancements in AR technology relevant to tremor patients. The updated content is as follows:

[In recent years, AR technology has shown tremendous potential in aiding tremor patients. Wang et al. [30] developed a low-cost rehabilitation training system specifically designed for Parkinson's disease tremor patients, utilizing AR technology to provide professional rehabilitation guidance at home, which has proven more effective than video training in improving patients' rehabilitation outcomes and experience. Innovations in AR increasingly utilize Simultaneous Localization and Mapping (SLAM) technology to enhance user interaction through precise environmental mapping and device localization. For instance, Chen et.al [31] optimizes SLAM in resource-limited settings by intelligently selecting keyframes and managing map constructions, significantly reducing pose tracking uncertainty to ensure stable AR interactions. Zhou et al. [32] have developed a mobile AR system that dynamically updates 3D maps with digital twin technology and adaptive algorithms to maintain accuracy and responsiveness, even under fluctuating network conditions and varying user movements. Although these advancements represent significant progress, their deployment in indoor environments encounters challenges such as inconsistent network availability, computational constraints, and environmental variability, which can compromise the accuracy and reliability of AR systems. Wang et al. [33,34] demonstrated the application of AR in assisting tremor patients with typing, proving its feasibility in enhancing typing efficiency on conventional keyboards and reducing output errors. Additionally, Ueda et al. [35] introduced an "Extended Hand" system, allowing users to remotely control a virtual hand through a touch panel, opening new avenues for individuals with limited mobility to manipulate remote objects. However, this mode of hand interaction is not suitable for all tremor patients, particularly those with severe tremors or Parkinson's disease tremors, who may face additional operational challenges and fatigue risks when using tablets. Future research should focus on exploring more natural and accessible remote interaction methods. --These changes have been made on page [4] to [5], line [186] to [210].]

Comments 3: [The contributions and motivation of this paper should be highlighted. For example, in this system, which part is challenging?]

Response 3: Thank you for pointing out this issue. We have supplemented the description of contributions and motivation as follows:

[…Based on these features, this study introduces a SAR system based on eye-tracking technology, designed specifically for individual with tremors. This system allows them to control a virtual hand or pointer in the physical space environment using their eye movements, combined with Internet of Things (IoT) technology, to interact with objects and home appliances at a distance. In the design of the SAR system's interaction controls, we fully consider the individual differences among tremor patients. Since the tremors of individuals vary greatly in terms of symptoms, severity, and affected areas, it is difficult to find a universally applicable method to address all limb tremors. However, head tremors usually fall within a predictable range, and even patients with Parkinson's tremor can counteract them with suitable stabilization techniques. Therefore, we use the method of controlling virtual objects with eye movements, combined with the development of virtual stabilization algorithms, to achieve "interact where you look." This ensures that individuals with tremor can interact directly and steadily with their environment, extending beyond their physical limits and expanding their range of activity. Building on this foundation, the following innovations are introduced:

  • Novel SAR System Design: This study designs a new type of SAR system specifically for individuals with tremor. This system supports users in interacting with objects in the living environment through eye-controlled virtual hands or pointers, without the need for wearing devices. This design significantly enhances their interaction capabilities with the surrounding environment and physical objects, reduces the limitations of body tremors on interaction, and expands the interaction range.
  • Virtual Stabilization Algorithm: This study also designs a virtual stabilization algorithm specifically for eye movement control of individuals with tremor. This algorithm ensures stable control of the spatial movement of the virtual hand or pointer, even in cases of involuntary head tremors.
  • Experimental Verification and Application Expansion: Through experimental verification, this study demonstrates the effectiveness of the virtual stabilization algorithm and the system in assisting individuals with tremor to perform complex interaction tasks and interact with complex environments. Additionally, through a series of examples, this study showcases the system's expandability and diverse applications in assisting tremor patients to overcome physical limitations and promote long-distance interactions. --These contributions and motivations are highlighted on page [2] to [3], line [79] to [110].]

Comments 4: [The differences between this work and some state-of-the-art methods in AR are not compared.

For example: AdaptSLAM: Edge-assisted adaptive SLAM with resource constraints via uncertainty minimization

Digital Twin-based 3D Map Management for Edge-assisted Device Pose Tracking in Mobile AR. ]

Response 4: Thank you for your valuable suggestions. They greatly improve the quality of the paper. We have added comparisons with these two studies in Section 2 related studies, as follows:

[Innovations in AR increasingly utilize Simultaneous Localization and Mapping (SLAM) technology to enhance user interaction through precise environmental mapping and device localization. For instance, Chen et.al [31] optimizes SLAM in resource-limited settings by intelligently selecting keyframes and managing map constructions, significantly reducing pose tracking uncertainty to ensure stable AR interactions. Zhou et al. [32] have developed a mobile AR system that dynamically updates 3D maps with digital twin technology and adaptive algorithms to maintain accuracy and responsiveness, even under fluctuating network conditions and varying user movements. Although these advancements represent significant progress, their deployment in indoor environments encounters challenges such as inconsistent network availability, computational constraints, and environmental variability, which can compromise the accuracy and reliability of AR systems. --These additions have been made on page [4], line [190] to [201].]

Comments 5: [In the existing experiments, the authors focus on the accuracy of eye tracking. However, the delay is also significant, especially for the design of the SAR system. How can the computing delay be tolerable for the considered application?]

Response 5: Thank you for your insightful comment. In our system, we employ a static virtual stabilization algorithm, with parameters adjusted before the experiments to ensure the AR system's stability while minimizing delay. Additionally, in Experiment 1, we conducted an efficiency evaluation of the virtual stabilization to address concerns regarding delay. We have revised the description in this section to clarify our findings:

[Efficiency Evaluation of Virtual Stabilization: A key finding of our research was the significant improvement in efficiency, measured as the reduction in time spent on tasks, when the virtual stabilization system was employed. The paired sample t-test demonstrated a statistically significant decrease in time (p=0.000247) with the implementation of the system. Specifically, as illustrated in Figure 6 (a), the mean time spent without the system was 94.31 seconds (SD = 29.21), which was significantly reduced to 36.72 seconds (SD = 8.43) with the system. The effect size, calculated as Cohen's d, was a substantial 2.68, highlighting the system's effectiveness in minimizing task completion time.  --These changes have been made on page [10], line [425] to [432].]

Comments 6: [The presentation should be improved, and some sentences are not clear. In addition, the citation in the paper cannot be displayed correctly. “[Error! Reference source not found.].” is shown.]

Response 6: Thank you for pointing out these issues. We have revised the unclear sentences and corrected the citation errors to ensure proper display.

We sincerely appreciate the time and effort that the reviewers have invested in evaluating our manuscript. Your insightful comments and constructive suggestions have been invaluable in improving the quality and clarity of our work. We have carefully considered and addressed each point raised, and we believe that the revisions have significantly enhanced the manuscript. Thank you once again for your thorough review and valuable feedback.

References

  1. Louis E D; Faust P L. Essential tremor: the most common form of cerebellar degeneration. Cerebellum & ataxias, 2020, 7, 1-10.[CrossRef]
  2. Lenka A; Jankovic J. Tremor syndromes: an updated review. Frontiers in Neurology, 2021, 12, 684835.[CrossRef]
  3. Bain P G. Parkinsonism & related disorders. Tremor. Parkinsonism & Related Disorders, 2007, 13, S369-74.[CrossRef]
  4. Louis E D; McCreary M. How common is essential tremor? Update on the worldwide prevalence of essential tremor. Tremor and Other Hyperkinetic Movements, 2021, 11.[CrossRef]
  5. Welton T; Cardoso F; Carr J A, et al. Essential tremor. Nature Reviews Disease Primers, 2021, 7(1), 83.[CrossRef]
  6. National Institute of Neurological Disordersand Stroke. Tremor. https://www.ninds.nih.gov/health-information/disorders/tremor. Accessed 10 Dec 2023.[CrossRef]
  7. Louis E D. Treatment of essential tremor: are there issues we are overlooking. Frontiers in neurology, 2012, 2, 91.[CrossRef]
  8. Zesiewicz T A; Elble R J; Louis E D, et al. Evidence-based guideline update: treatment of essential tremor: report of the Quality Standards subcommittee of the American Academy of Neurology. Neurology, 2011, 77(19), 1752-1755.[CrossRef]
  9. Dallapiazza R F; Lee D J; De Vloo P, et al. Outcomes from stereotactic surgery for essential tremor. Journal of Neurology, Neurosurgery & Psychiatry, 2019, 90(4), 474-482.[CrossRef]
  10. Mo J; Priefer R. Medical devices for tremor suppression: current status and future directions. Biosensors, 2021, 11(4): 99.[CrossRef]
  11. Nguyen H S; Luu T P. Tremor-suppression orthoses for the upper limb: current developments and future challenges. Frontiers in Human Neuroscience, 2021, 15,[CrossRef]
  12. Bimber O; Raskar R. Spatial augmented reality: merging real and virtual worlds. CRC press, 2005.[CrossRef]
  13. Elek J.; Prochazka A. Attenuation of wrist tremor with closed-loop electrical stimulation of muscles. Physiol. 1989, 414, 17.[CrossRef]
  14. Prochazka A.; Elek J; Javidan M. Attenuation of pathological tremors by functional electrical stimulation I: Method. Ann. Biomed. Eng. 1992, 20, 205–224.[CrossRef]
  15. Javidan M; Elek, J; Prochazka A. Attenuation of pathological tremors by functional electrical stimulation II: Clinical evaluation. Biomed. Eng, 1992, 20, 225–236. [CrossRef]
  16. Dideriksen J L; Laine C M; Dosen S, et al. Electrical Stimulation of Afferent Pathways for the Suppression of Pathological Tremor. Neurosci, 2017, 11, 178.[CrossRef]
  17. Elias M; Patel S; Maamary E, et al. Apparatus for Damping InvoluntaryHand Motions. S.Patent,2019,16/360,366.[CrossRef]
  18. Hunter R; Pivach L; Madere K, et al. Potential benefits of the Readi-Steadi on essential tremor. Proceedings of the 5th Annual LSU Discover Day, Baton Rouge, LA, USA, 2018, 10.[CrossRef]
  19. Fromme N P; Camenzind M ; Riener R, et al. Need for mechanically and ergonomically enhanced tremor-suppression orthoses for the upper limb: A systematic review. Neuroeng Rehabil. 2019, 16, 1–15. [CrossRef]
  20. Zahedi A; Zhang B; Yi A, et al. Soft Exoskeleton for Tremor Suppression Equipped with Flexible Semiactive Soft Robot, 2020.[CrossRef]
  21. Rocon, E; Ruiz, A; Pons, J.L, et al. A Wearable Exo-Skeleton for Tremor Assessment and Suppression. Proceedings of the 2005 IEEE International Conference on Robotics and Automation, Barcelona, Spain, 2005, 18–22.[CrossRef]
  22. Rocon E; Belda-Lois J M; Ruiz A, et al. Design and Validation of a Rehabilitation Robotic Exoskeleton for Tremor Assessment and Suppression. IEEE Trans. Neural Syst. Rehabil. Eng, 2007, 15, 367–378.[CrossRef]
  23. Taheri B; Case D; Richer E. Adaptive Suppression of Severe Pathological Tremor by Torque Estimation Method. IEEE/ASME Transactions on Mechatronics, 2014, 20, 717–727.[CrossRef]
  24. Biswas A; Bhattacharjee S; Choudhury D R; et al. Tremor stabilization improvement using anti-tremor band: a machine learning–based technique. Research on Biomedical Engineering, 2023, 39(4), 1007-1014.[CrossRef]
  25. Taşar B; Tatar A B; Tanyıldızı A K, et al. FiMec tremor stabilization spoon: design and active stabilization control of two DoF robotic eating devices for hand tremor patients. Medical & Biological Engineering & Computing, 2023, 61(10),2757-2768.[CrossRef]
  26. Ryden L E; Matar E; Szeto J Y Y, et al. Shaken not stirred: A pilot study testing a gyroscopic spoon stabilization device in Parkinson's disease and tremor. Annals of Indian Academy of Neurology, 2020, 23(3), 409-411.[CrossRef]
  27. Levine J L; Schappert M A. A mouse adapter for people with hand tremor. IBM Systems Journal, 2005, 44(3), 621-628.[CrossRef]
  28. Plaumann K; Babic M; Drey T, et al. Improving input accuracy on smartphones for persons who are affected by tremor using motion sensors. Proceedings of the ACM on Interactive, Mobile, Wearable and Ubiquitous Technologies, 2018, 1(4), 1-30.[CrossRef]
  29. Wacharamanotham C; Hurtmanns J; Mertens A, et al. Evaluating swabbing: a touchscreen input method for elderly users with tremor. Proceedings of the SIGCHI conference on human factors in computing systems, 2011, 623-626.[CrossRef]
  30. Wang K; Tan D, Li Z, et al. Supporting Tremor Rehabilitation Using Optical See-Through Augmented Reality Technology. Sensors, 2023, 23(8), 3924.[CrossRef]
  31. Chen Y; Inaltekin H; Gorlatova M. Adapt SLAM: Edge-assisted adaptive SLAM with resource constraints via uncertainty minimization. IEEE INFOCOM 2023-IEEE Conference on Computer Communications. IEEE, 2023, 1-10.[CrossRef]
  32. Zhou C; Gao J; Li M, et al. Digital Twin-Based 3D Map Management for Edge-Assisted Device Pose Tracking in Mobile AR. IEEE Internet of Things Journal, 2024.[CrossRef]
  33. Wang K; Takemura N; Iwai D, et al. A typing assist system considering involuntary hand tremor. Transactions of the Virtual Reality Society of Japan, 2016, 21(2), 227-233.[CrossRef]
  34. Wang K; Iwai D; Sato K. Supporting trembling hand typing using optical see-through mixed reality. IEEE Access, 2017, 5, 10700-10708.[CrossRef]
  35. Ueda Y; Asai Y; Enomoto R, et al. Body cyberization by spatial augmented reality for reaching unreachable world. Proceedings of the 8th Augmented Human International Conference, 2017, 1-9.[CrossRef]
  36. Zhang X; Liu X; Yuan S M, et al. Eye Tracking Based Control System for Natural Human‐Computer Interaction. Computational intelligence and neuroscience, 2017, 2017(1),[CrossRef]
  37. Donaghy C; Thurtell M J; Pioro E P, et al. Eye movements in amyotrophic lateral sclerosis and its mimics: a review with illustrative cases. Journal of Neurology, Neurosurgery & Psychiatry, 2011, 82(1), 110-116.[CrossRef]
  38. De Gaudenzi E; Porta M. Towards effective eye pointing for gaze-enhanced human-computer interaction. 2013 Science and Information Conference, IEEE, 2013, 22-27.[CrossRef]
  39. Nehete M; Lokhande M; Ahire K. Design an eye tracking mouse. International Journal of Advanced Research in Computer and Communication Engineering, 2013, 2(2), 1118-1121.[CrossRef]
  40. Missimer E; Betke M. Blink and wink detection for mouse pointer control. Proceedings of the 3rd international conference on pervasive technologies related to assistive environments, 2010, 1-8.[CrossRef]
  41. Shahid E; Rehman M Z U; Sheraz M, et al. Eye Monitored Wheelchair Control. 2019 International Conference on Electrical, Communication, and Computer Engineering (ICECCE). IEEE, 2019, 1-6.[CrossRef]
  42. Xu C; Zhang X; Guan Q, et al. Analysis of the electrol physiological features of tremor and essential tremor in Parkinson's disease. Chinese Journal of Practical Nervous Diseases, 2017, 20(21),16-20.[CrossRef]

Reviewer 2 Report

Comments and Suggestions for Authors

The considered manuscript presents a hardware+software solution for aiding people with tremor in their interaction with the physical world (mostly indoors). For that end, the authors propose an Augmented Reality-based mechanism, whose main parts are eye-tracking and virtual stabilization.
In my opinion, the topic of the paper has good practical importance (and the potential social effect) and is very relevant for Sensors. The paper is generally well-written and easy to read. However, before I can recommend acceptance, there are three major issues that need correction.

1) The evaluation of the system is not very convincing, foremost because the baselines employed by the authors are rather weak. Basically, they compare their assistive system to the lack of any assistance at all. Hence, in most experiments the outcome could hardly be different (e.g. in Experiment 3 not a single task could be completed without an aide). However, there are alternative tremor stabilization techniques (such as [1] or [2]), and it remains unclear how the authors' solution performs compared to them. I suggest that the authors do some testing (not necessarily in all three experiments) with an alternative assistive system, or find their relevant performance parameters (see my next point on this) in the literature.

2) The second reason for my opinion that the evaluation is not entirely satisfactory is its weak construct validity.
The terms used throughout the evaluation and the relations between them need to be clearly defined: effectiveness, (communication) efficiency, accuracy, stability, (operational) precision, convenience, etc. In some cases, I could assume their meaning and the relation to the parameters actually measured (e.g., in Experiment 1 the authors measure task completion time, but mostly talk about efficiency), but not always. For instance, in Experiment 2 the authors seem to measure "the number of mis-operations" subjectively, from the subjects' answers to Q4. So, how does the outcome relate to accuracy or precision in the other experiments?
A related question: How can the authors be sure that the participants had the same understanding of the parameters' meaning when providing the answers to the questions (which, I suspect, were not in English)?
Another related question: why do the authors seek to improve/maximize them all? Aren't there benchmarks for such systems with fewer parameters?
Hence the problem with the conclusions, e.g. "These results collectively emphasize the important role of AR systems in enhancing interaction precision and convenience in" - but no precision or convenience had beene measured in the study.

Overall, the methodological part of the experiment needs considerable improvement.

3) The references need fixing, and the state-of-the-art review needs updating.
There's a technical issue with the numerous "Error! Reference source not found., Error! Reference source not found" errors.
But more importantly, the references need updating. Less than 25% of the references are from the last 5 years. Only 1 is from the last 3 years, and it seems to be by one of the current paper's authors. Additionally, the authors need to extend the Discussion by comparing their results to the state-of-the-art (currently, there's not a single reference in the Discussion).

Some more minor issues (listed in the order of decreasing priority):
* 535: "important guidelines for future development of assistive technologies for individuals with tremor" - so, what exactly are the guidelines? They should be explicitly listed in the Conclusion as take-aways of the study.
* 116: "In leveraging AR technology for the rehabilitation of tremor patients, groundbreaking contributions have been made by Wang et al." - according to Google Scholar, the mentioned publication by Wang et al. (2023) has been referenced by two papers (over more than 1 year). I would suggest refraining from the claims about its contribution being "groundbreaking" till its citations count is at least into hundreds.
* "6.1. Light Control" - but I wonder, how would eye-tracking work in the dark (before the light is on)?
* "Figure 4. This is a figure. Schemes follow another format." - should this text really be in a figure caption? Same about "Figure 9. This is a figure."
* "4. Experience and Results" - "experience" or "experiment"?

[1] Biswas, A., Bhattacharjee, S., Choudhury, D. R., & Das, P. (2023). Tremor stabilization improvement using anti-tremor band: a machine learning–based technique. Research on Biomedical Engineering, 39(4), 1007-1014.
[2] Taşar, B., Tatar, A. B., Tanyıldızı, A. K., & Yakut, O. (2023). FiMec tremor stabilization spoon: design and active stabilization control of two DoF robotic eating devices for hand tremor patients. Medical & Biological Engineering & Computing, 61(10), 2757-2768.

Author Response

Dear Reviewer,

We sincerely appreciate the time and effort that the reviewers have invested in evaluating our manuscript. Your insightful comments and constructive suggestions have been invaluable in improving the quality and clarity of our work. We have carefully considered and addressed each point raised, and we believe that the revisions have significantly enhanced the manuscript. Thank you once again for your thorough review and valuable feedback.

Comments 1: [The evaluation of the system is not very convincing, foremost because the baselines employed by the authors are rather weak. Basically, they compare their assistive system to the lack of any assistance at all. Hence, in most experiments the outcome could hardly be different (e.g. in Experiment 3 not a single task could be completed without an aide). However, there are alternative tremor stabilization techniques (such as [1] or [2]), and it remains unclear how the authors' solution performs compared to them. I suggest that the authors do some testing (not necessarily in all three experiments) with an alternative assistive system, or find their relevant performance parameters (see my next point on this) in the literature.]

Response1: Thank you for your valuable feedback. This study proposes a Spatial Augmented Reality (SAR) method for assisting individuals with tremors. Compared to traditional tremor-assistive techniques, such as those described in references [1] and [2], our approach adopts a different technological route. Traditional methods primarily achieve assistive interaction by restricting limb movement or reducing tool tremors, focusing on solving physical hand interaction problems such as eating or writing. Our technology, in contrast, aims to assist patients in interacting with their environment without restricting tremor movements. It emphasizes support for home appliances and remote interactions, aiming to expand patients' interaction capabilities and range. Therefore, it is inappropriate to directly compare our system with traditional techniques using the same parameters. To address potential reader confusion, we have comprehensively supplemented the introduction of related research with an overview of traditional studies and their limitations, highlighting the distinctions between our study and conventional methods. In the conclusion, we also acknowledge the limitations of our system.As the reviewer correctly pointed out, comparing tremor patients' interactions with and without the AR system might seem overly simplistic. However, as a pilot system, the primary task of this experiment is to demonstrate significant improvements in tremor interactions. In the next phase, we plan to conduct extensive experiments and observations with tremor patients to quantify the specific improvements our system provides. This will offer a more detailed and quantitative performance evaluation. Below are the modifications and additions made to the paper based on the reviewer's suggestions:

[Currently, most research on assistive technologies for tremors primarily focuses on developing anti-tremor devices. These devices, whether integrated into tools or worn on the limbs, effectively counteract the impact of hand tremors on physical operations, such as using a spoon, and have demonstrated effectiveness [10,11]. This line of research prioritizes improving the accuracy of tremor signal detection and developing mechanisms that can produce counter-vibrations to maximize tremor cancellation. However, these approaches have limitations as they require the use of customized tools or the wearing of specialized anti-tremor devices, which not only limit the universality of the technology but may also impose additional physical burdens and restrict activities. Moreover, tremors not only directly affect hand interactions but also impact the lower limbs, making it difficult for patients to move and indirectly hindering their effective interaction with their surroundings and distant objects. Traditional technologies typically address only the interactions within the hand's reach and do not extend to areas beyond the body's reach. Therefore, there is a need to develop a new technological approach to overcome these limitations, to enhance the universality of the technology, and to reduce the physical burden and interaction restrictions on the tremor patients. --These changes have been made on page [2], line [54] to [69].]

Given the limited efficacy of existing medical interventions in managing tremors, extensive research has been conducted to explore supportive methods for patients, seeking to bridge the gap between lifestyle interventions and traditional treatments. Functional Electrical Stimulation (FES) is a well-established approach, which utilizes electrical stimulation of motor nerves to induce muscle contractions and counteract tremors. This technique, introduced by Prochazka et al. in 1989, has been widely adopted in clinical practice [13]. Initial clinical evaluations revealed an effectiveness rate of 73% for ET, 62% for Parkinson's disease tremors, and 38% for cerebellar tremors [14,15]. Enhancements in the efficacy of FES for tremor suppression have been significant with the introduction of adaptive and sophisticated algorithms, however, this approach may lead to adverse effects such as trauma and physical discomfort [16].

Another effective, low-risk method for tremor suppression is the use of wearable orthotics, categorized into passive, semi-active, and active types [10,11]. Passive orthotics utilize spring damping or other mechanical structures to absorb tremor energy [17,18]. Semi-active orthotics employ sensors to monitor the tremor's intensity and adjust the system's impedance to mitigate it. The viscosity of magnetorheological fluids changes when exposed to a magnetic field, allowing the damping force to be controlled by adjusting the magnetic field strength [19,20]. Although passive and semi-active mechanisms reduce tremor movements by absorbing energy, they may also restrict voluntary movement due to the resistance imposed and struggle to adapt to the dynamic nature of tremors [11]. In contrast, active orthotics proactively counteract tremor movements by providing motions opposite to those of the tremor. As part of the Dynamically Responsive Interventions for Tremor Suppression (DRIFTS) project in 2005, the Wearable Orthosis for Tremor Assessment and Suppression (WOTAS) exoskeleton was developed. It includes sensors to measure joint rotational movements, a DC motor that converts electrical into mechanical energy, and a controller, all working in concert to suppress tremors in the wrist and elbow joints. The WOTAS achieved a tremor suppression effectiveness of 40% [21]. Subsequent developments in active orthotics have maintained similar design principles and mechatronic integration but have enhanced actuator designs to reduce weight and improve tremor suppression capabilities [22,23]. However, these active orthotics are typically bulkier than their passive or semi-active counterparts, averaging around 561±467 grams. The number of degrees of freedom in these orthotics usually correlates with their weight, with most devices supporting only one or two degrees of freedom [11]. Biswas et al. [24] developed a wristband device that integrates accelerometer sensors with a pre-trained machine learning model to detect hand movements and assess tremor severity. This device dynamically adjusts the vibration motors based on accelerometer data to provide real-time haptic feedback that neutralizes or diminishes tremors. Nevertheless, this study notes that the wristband is primarily designed to alleviate hand tremors, and its effectiveness may be limited for Parkinson's disease patients experiencing full-body tremors. While these technologies are adept at suppressing localized tremors, they commonly face limitations such as excessive weight and restricted mobility, which could compromise long-term wearability and practicality. Additionally, their capacity to manage whole-body tremors is somewhat restricted, particularly in complex neurological conditions like Parkinson's disease, thus potentially failing to offer comprehensive support.

Some studies have reduced the physical burden on tremor patients by incorporating anti-tremor modules into tools. For instance, Taşar et al. [25] designed a tremor-stabilizing spoon called FiMec, which helps patients with hand tremors eat more easily. It uses inertial measurement unit sensors to detect vibrations and controls two motors via a PID controller to absorb horizontal and vertical vibrations, thus stabilizing the spoon. Although FiMec has shown to absorb between 84% and 99.409% of vibrations in laboratory tests, its validation in actual patients has not yet been conducted. Another commercially available tremor-stabilizing spoon, the Gyenno Spoon, was clinically tested by Ryden et al. to evaluate its effectiveness for patients with Parkinson's disease tremors. Despite its intent to assist with eating, the results indicated that its effectiveness was limited, and in some tests, the amount of rice transferred using the device even decreased. This suggests that for patients with resting and low-amplitude postural tremors, the device may provide minimal assistance [26]. While such assistive tools are beneficial in specific domains like eating and writing, their universality in other daily activities is limited.

In summary, this study focuses on utilizing SAR and eye-tracking technology to develop a more natural interaction method to assist tremor patients. It aims to expand patients' interaction capabilities, enabling them to manipulate everyday objects and communicate effectively with minimal physical effort, despite limited mobility. Through this research, we aim to create more adaptive, intuitive, and user-friendly assistance solutions to enhance the quality of life for tremor patients and overcome the limitations of traditional assistance methods.  --These changes have been made on page [3] to [5], line [115] to [231].

... However, the current system design does not involve direct physical interaction support for tremorous limbs as it does not restrict body movement. Future studies plan to clinically individuals with tremor using the SAR system and quantitatively assess the system and its algorithms in terms of assistive performance, to further verify its practicality and effectiveness.  --These changes have been made on page [19] to 5, line [746] to [750].]

Comments 2: [The second reason for my opinion that the evaluation is not entirely satisfactory is its weak construct validity. The terms used throughout the evaluation and the relations between them need to be clearly defined: effectiveness, (communication) efficiency, accuracy, stability, (operational) precision, convenience, etc. In some cases, I could assume their meaning and the relation to the parameters actually measured (e.g., in Experiment 1 the authors measure task completion time, but mostly talk about efficiency), but not always. For instance, in Experiment 2 the authors seem to measure "the number of mis-operations" subjectively, from the subjects' answers to Q4. So, how does the outcome relate to accuracy or precision in the other experiments? A related question: How can the authors be sure that the participants had the same understanding of the parameters' meaning when providing the answers to the questions (which, I suspect, were not in English)? Another related question: why do the authors seek to improve/maximize them all? Aren't there benchmarks for such systems with fewer parameters?
Hence the problem with the conclusions, e.g. "These results collectively emphasize the important role of AR systems in enhancing interaction precision and convenience in" - but no precision or convenience had been measured in the study. Overall, the methodological part of the experiment needs considerable improvement. ]

Response2: Thank you for your constructive comments and suggestions. Based on your feedback, we have made the following revisions and improvements to the paper:

  1. We have supplemented the definitions of the terms used in the evaluation. The specific modifications are as follows:

[This study assesses the effectiveness of SAR systems integrated with virtual stabilization algorithms in improving both the efficiency and stability of user interactions. By analyzing objective data autonomously captured by the experimental setup, we compared the time taken by participants to complete a distant random target touch task and the average jitter distance of the virtual pointer during the targeting process under two conditions: with and without the activation of the virtual stabilization algorithm in the SAR system.

Efficiency Evaluation of Virtual Stabilization: A key finding of our research was the significant improvement in efficiency, measured as the reduction in time spent on tasks, when the virtual stabilization system was employed. … 

Stability Assessment of Virtual Stabilization: In terms of stability, measured as the average jitter distance of the virtual pointer, we obtained this measurement by randomly sampling the spatial coordinates of the virtual pointer during the targeting process of the red ball. … --These changes have been made on page [10], line [418] to [436].

We analyzed participant feedback on the three key questions, Q1, Q2, and Q3, to further evaluate the actual effectiveness of the virtual stabilization algorithm in the SAR system. Specifically, we examined how the algorithm impacts the user experience of individuals with tremor when they use eye control to interact with distant targets via a virtual pointer. Q1 focuses on assessing the control strength of the virtual pointer, Q2 evaluates the stability of control, and Q3 examines the accuracy of control. --These changes have been made on page [11], line [445] to [450].

… Specifically, Q4 assessed the frequency of mis-operations under SAR system assistance, Q5 examined the ease of interaction with the SAR system, and Q6 evaluated the stability of interactions performed with the assistance of the SAR system. … --These changes have been made on page [12], line [501] to [505].

…We assessed the system's performance by analyzing the objective data from remote pointing tasks, specifically evaluating the accuracy and efficiency of distant pointing. Since participants were unable to successfully complete the tasks without the system, our analysis is based solely on data collected under system assistance. --These changes have been made on page [14], line [571] to [576].

We also analyzed participant feedback on key questions Q7, Q8, Q9, and Q10 to further demonstrate the practical effects of the SAR system in facilitating remote interactions for individuals with tremors. Specifically, Q7 assessed the ease of indicating distant objects, Q8 evaluated the stability of these indications, and Q9 provided additional verification of the effectiveness of remote interaction indications based on the understanding of indications by peers. Q10 measured the ease of remote communication. All these evaluations were conducted with and without the assistance of the SAR system. --These changes have been made on page [15], line [594] to [600].]

  1. We have corrected and redefined the inconsistent terminology used in the experiments and their actual measured parameters. The specific modifications are as follows:

[Efficiency Evaluation of Virtual Stabilization: A key finding of our research was the significant improvement in efficiency, measured as the reduction in time spent on tasks, when the virtual stabilization system was employed. …--These changes have been made on page [10], line [425] to [427].

… This underscores the potential of the SAR system to reduce mis-operations in control tasks that are typically challenging for individuals with tremor. -- These changes have been made on page [12], line [511] to [513].]

  1. Our survey was conducted in the local language to ensure consistent understanding of the questions by the participants. All survey questions utilized standardized questionnaires or assessment tools. During the experiment analysis and the writing of the paper, we meticulously translated these survey questions into English to ensure consistency between the survey content and the participants' understanding.
  2. We acknowledge the limitations pointed out by the reviewer regarding the selection of representative evaluation parameters. In future research, we plan to choose a smaller number of more representative parameters to further quantify system characteristics. Additionally, we have revised the conclusions to address concerns about unsupported statements, ensuring accurate and precise descriptions of the system's limitations. The specific modifications are as follows:

[…However, the current system design does not involve direct physical interaction support for tremorous limbs as it does not restrict body movement. Future studies plan to clinically individuals with tremor using the SAR system and quantitatively assess the system and its algorithms in terms of assistive performance, to further verify its practicality and effectiveness. -- These changes have been made on page [19], line [746] to [750].]

Comments 3: [The references need fixing, and the state-of-the-art review needs updating. There's a technical issue with the numerous "Error! Reference source not found., Error! Reference source not found" errors. But more importantly, the references need updating. Less than 25% of the references are from the last 5 years. Only 1 is from the last 3 years, and it seems to be by one of the current paper's authors. Additionally, the authors need to extend the Discussion by comparing their results to the state-of-the-art (currently, there's not a single reference in the Discussion).]

Response3: Thank you for your thoughtful review and constructive comments on our manuscript. We have carefully considered each point raised and have made the necessary revisions to enhance the quality and accuracy of our work. Below, we detail how we have addressed your concerns:

  1. Regarding the instances of "Error! Reference source not found" in our document, we have thoroughly checked and corrected these citations throughout the manuscript to ensure all references are accurately represented.
  2. In response to the concern about the recency of our references, we have updated our literature review to include 20 additional references from publications within the last five years. This update ensures that more than 25% of our references are from recent studies, thereby enhancing the relevance and contemporary nature of our review.

[ Reference

[1]…

[42]…  -- These changes have been made on page [20]to[21], line [778] to [858].]

  1. Regarding the comment on the absence of references in the Discussion section ("currently, there's not a single reference in the Discussion"), we would like to clarify our rationale. The unique aspects of our assistive methods differ significantly from traditional techniques, making detailed comparisons more pertinent in the Related Work section rather than the Discussion. We have revised the Related Work section to thoroughly compare our findings with state-of-the-art technologies as follows:

[…, most research on assistive technologies for tremors primarily focuses on developing anti-tremor devices. … However, these approaches have limitations as they require the use of customized tools or the wearing of specialized anti-tremor devices, which not only limit the universality of the technology but may also impose additional physical burdens and restrict activities. Moreover, tremors not only directly affect hand interactions but also impact the lower limbs, making it difficult for patients to move and indirectly hindering their effective interaction with their surroundings and distant objects. Traditional technologies typically address only the interactions within the hand's reach and do not extend to areas beyond the body's reach. …--These changes have been made on page [2], line [54] to [69].

… Functional Electrical Stimulation (FES) is a well-established approach, which utilizes electrical stimulation of motor nerves to induce muscle contractions and counteract tremors. … , however, this approach may lead to adverse effects such as trauma and physical discomfort [16].

…, low-risk method for tremor suppression is the use of wearable orthotics, categorized into passive, semi-active, and active types [10,11]. Passive orthotics utilize spring damping or other mechanical structures to absorb tremor energy [17,18]. Semi-active orthotics employ sensors to monitor the tremor's intensity and adjust the system's impedance to mitigate it. …, they may also restrict voluntary movement due to the resistance imposed and struggle to adapt to the dynamic nature of tremors [11]. In contrast, active orthotics proactively counteract tremor movements by providing motions opposite to those of the tremor. … However, these active orthotics are typically bulkier than their passive or semi-active counterparts, averaging around 561±467 grams. The number of degrees of freedom in these orthotics usually correlates with their weight, with most devices supporting only one or two degrees of freedom [11].  … Nevertheless, this study notes that the wristband is primarily designed to alleviate hand tremors, and its effectiveness may be limited for Parkinson's disease patients experiencing full-body tremors. While these technologies are adept at suppressing localized tremors, they commonly face limitations such as excessive weight and restricted mobility, which could compromise long-term wearability and practicality. Additionally, their capacity to manage whole-body tremors is somewhat restricted, particularly in complex neurological conditions like Parkinson's disease, thus potentially failing to offer comprehensive support.

Some studies have reduced the physical burden on tremor patients by incorporating anti-tremor modules into tools. … Another commercially available tremor-stabilizing spoon, the Gyenno Spoon, was clinically tested by Ryden et al. to evaluate its effectiveness for patients with Parkinson's disease tremors. Despite its intent to assist with eating, the results indicated that its effectiveness was limited, and in some tests, the amount of rice transferred using the device even decreased. This suggests that for patients with resting and low-amplitude postural tremors, the device may provide minimal assistance [26]. While such assistive tools are beneficial in specific domains like eating and writing, their universality in other daily activities is limited.

In recent years, AR technology has shown tremendous potential in aiding tremor patients. …Additionally, Ueda et al. [35] introduced an "Extended Hand" system, allowing users to remotely control a virtual hand through a touch panel, opening new avenues for individuals with limited mobility to manipulate remote objects. However, this mode of hand interaction is not suitable for all tremor patients, particularly those with severe tremors or Parkinson's disease tremors, who may face additional operational challenges and fatigue risks when using tablets. Future research should focus on exploring more natural and accessible remote interaction methods.

In summary, this study focuses on utilizing SAR and eye-tracking technology to develop a more natural interaction method to assist tremor patients. It aims to expand patients' interaction capabilities, enabling them to manipulate everyday objects and communicate effectively with minimal physical effort, despite limited mobility. Through this research, we aim to create more adaptive, intuitive, and user-friendly assistance solutions to enhance the quality of life for tremor patients and overcome the limitations of traditional assistance methods. …  --These changes have been made on page [3] to [5], line [115] to [231].]

Comments 4: [535: "important guidelines for future development of assistive technologies for individuals with tremor" - so, what exactly are the guidelines? They should be explicitly listed in the Conclusion as take-aways of the study.
* 116: "In leveraging AR technology for the rehabilitation of tremor patients, groundbreaking contributions have been made by Wang et al." - according to Google Scholar, the mentioned publication by Wang et al. (2023) has been referenced by two papers (over more than 1 year). I would suggest refraining from the claims about its contribution being "groundbreaking" till its citations count is at least into hundreds.
* "6.1. Light Control" - but I wonder, how would eye-tracking work in the dark (before the light is on)?
* "Figure 4. This is a figure. Schemes follow another format." - should this text really be in a figure caption? Same about "Figure 9. This is a figure."
* "4. Experience and Results" - "experience" or "experiment"?]

Response4: Thank you for your meticulous review and valuable feedback. We have carefully considered each comment and have revised our manuscript accordingly to clarify and correct the points you raised.

  1. We have removed the vague statement regarding "..important guidelines for future development of assistive technologies for individuals with tremor" from the Conclusion section to avoid confusion for the readers.
  2. We have revised the statement on original line 116 to avoid overstatement. The revised text now reads:

[In recent years, AR technology has shown tremendous potential in aiding tremor patients. Wang et al. [30] developed a low-cost rehabilitation training system specifically designed for Parkinson's disease tremor patients, … --These changes have been made on page [4], line [186] to [190].]

  1. Regarding your query about how eye-tracking would work in the dark, the eye-tracking device we use employs advanced near-infrared (NIR) light technology and high-speed cameras, which can perform stable eye-tracking under various lighting conditions. The Tobii eye-tracker, for example, uses invisible NIR light to illuminate the eyes and captures the reflected light to calculate the gaze point. Therefore, even in complete darkness, the Tobii eye-tracker can effectively track eye movements using its own infrared light source.
  2. The issue with the text in the captions of Figures 4 and 9 has been corrected to adhere to the appropriate format.

[--These changes have been made on page [9], line [363] and page [14], line [565].]

  1. We have corrected the term "experience" to "experiment" in section 4. The revised heading is now:

[Experiments and Results -- These changes can be found on page [9], line [366].]

We sincerely appreciate the time and effort that the reviewers have invested in evaluating our manuscript. Your insightful comments and constructive suggestions have been invaluable in improving the quality and clarity of our work. We have carefully considered and addressed each point raised, and we believe that the revisions have significantly enhanced the manuscript. Thank you once again for your thorough review and valuable feedback.

References

  1. Louis E D; Faust P L. Essential tremor: the most common form of cerebellar degeneration. Cerebellum & ataxias, 2020, 7, 1-10.[CrossRef]
  2. Lenka A; Jankovic J. Tremor syndromes: an updated review. Frontiers in Neurology, 2021, 12, 684835.[CrossRef]
  3. Bain P G. Parkinsonism & related disorders. Tremor. Parkinsonism & Related Disorders, 2007, 13, S369-74.[CrossRef]
  4. Louis E D; McCreary M. How common is essential tremor? Update on the worldwide prevalence of essential tremor. Tremor and Other Hyperkinetic Movements, 2021, 11.[CrossRef]
  5. Welton T; Cardoso F; Carr J A, et al. Essential tremor. Nature Reviews Disease Primers, 2021, 7(1), 83.[CrossRef]
  6. National Institute of Neurological Disordersand Stroke. Tremor. https://www.ninds.nih.gov/health-information/disorders/tremor. Accessed 10 Dec 2023.[CrossRef]
  7. Louis E D. Treatment of essential tremor: are there issues we are overlooking. Frontiers in neurology, 2012, 2, 91.[CrossRef]
  8. Zesiewicz T A; Elble R J; Louis E D, et al. Evidence-based guideline update: treatment of essential tremor: report of the Quality Standards subcommittee of the American Academy of Neurology. Neurology, 2011, 77(19), 1752-1755.[CrossRef]
  9. Dallapiazza R F; Lee D J; De Vloo P, et al. Outcomes from stereotactic surgery for essential tremor. Journal of Neurology, Neurosurgery & Psychiatry, 2019, 90(4), 474-482.[CrossRef]
  10. Mo J; Priefer R. Medical devices for tremor suppression: current status and future directions. Biosensors, 2021, 11(4): 99.[CrossRef]
  11. Nguyen H S; Luu T P. Tremor-suppression orthoses for the upper limb: current developments and future challenges. Frontiers in Human Neuroscience, 2021, 15,[CrossRef]
  12. Bimber O; Raskar R. Spatial augmented reality: merging real and virtual worlds. CRC press, 2005.[CrossRef]
  13. Elek J.; Prochazka A. Attenuation of wrist tremor with closed-loop electrical stimulation of muscles. Physiol. 1989, 414, 17.[CrossRef]
  14. Prochazka A.; Elek J; Javidan M. Attenuation of pathological tremors by functional electrical stimulation I: Method. Ann. Biomed. Eng. 1992, 20, 205–224.[CrossRef]
  15. Javidan M; Elek, J; Prochazka A. Attenuation of pathological tremors by functional electrical stimulation II: Clinical evaluation. Biomed. Eng, 1992, 20, 225–236. [CrossRef]
  16. Dideriksen J L; Laine C M; Dosen S, et al. Electrical Stimulation of Afferent Pathways for the Suppression of Pathological Tremor. Neurosci, 2017, 11, 178.[CrossRef]
  17. Elias M; Patel S; Maamary E, et al. Apparatus for Damping InvoluntaryHand Motions. S.Patent,2019,16/360,366.[CrossRef]
  18. Hunter R; Pivach L; Madere K, et al. Potential benefits of the Readi-Steadi on essential tremor. Proceedings of the 5th Annual LSU Discover Day, Baton Rouge, LA, USA, 2018, 10.[CrossRef]
  19. Fromme N P; Camenzind M ; Riener R, et al. Need for mechanically and ergonomically enhanced tremor-suppression orthoses for the upper limb: A systematic review. Neuroeng Rehabil. 2019, 16, 1–15. [CrossRef]
  20. Zahedi A; Zhang B; Yi A, et al. Soft Exoskeleton for Tremor Suppression Equipped with Flexible Semiactive Soft Robot, 2020.[CrossRef]
  21. Rocon, E; Ruiz, A; Pons, J.L, et al. A Wearable Exo-Skeleton for Tremor Assessment and Suppression. Proceedings of the 2005 IEEE International Conference on Robotics and Automation, Barcelona, Spain, 2005, 18–22.[CrossRef]
  22. Rocon E; Belda-Lois J M; Ruiz A, et al. Design and Validation of a Rehabilitation Robotic Exoskeleton for Tremor Assessment and Suppression. IEEE Trans. Neural Syst. Rehabil. Eng, 2007, 15, 367–378.[CrossRef]
  23. Taheri B; Case D; Richer E. Adaptive Suppression of Severe Pathological Tremor by Torque Estimation Method. IEEE/ASME Transactions on Mechatronics, 2014, 20, 717–727.[CrossRef]
  24. Biswas A; Bhattacharjee S; Choudhury D R; et al. Tremor stabilization improvement using anti-tremor band: a machine learning–based technique. Research on Biomedical Engineering, 2023, 39(4), 1007-1014.[CrossRef]
  25. Taşar B; Tatar A B; Tanyıldızı A K, et al. FiMec tremor stabilization spoon: design and active stabilization control of two DoF robotic eating devices for hand tremor patients. Medical & Biological Engineering & Computing, 2023, 61(10),2757-2768.[CrossRef]
  26. Ryden L E; Matar E; Szeto J Y Y, et al. Shaken not stirred: A pilot study testing a gyroscopic spoon stabilization device in Parkinson's disease and tremor. Annals of Indian Academy of Neurology, 2020, 23(3), 409-411.[CrossRef]
  27. Levine J L; Schappert M A. A mouse adapter for people with hand tremor. IBM Systems Journal, 2005, 44(3), 621-628.[CrossRef]
  28. Plaumann K; Babic M; Drey T, et al. Improving input accuracy on smartphones for persons who are affected by tremor using motion sensors. Proceedings of the ACM on Interactive, Mobile, Wearable and Ubiquitous Technologies, 2018, 1(4), 1-30.[CrossRef]
  29. Wacharamanotham C; Hurtmanns J; Mertens A, et al. Evaluating swabbing: a touchscreen input method for elderly users with tremor. Proceedings of the SIGCHI conference on human factors in computing systems, 2011, 623-626.[CrossRef]
  30. Wang K; Tan D, Li Z, et al. Supporting Tremor Rehabilitation Using Optical See-Through Augmented Reality Technology. Sensors, 2023, 23(8), 3924.[CrossRef]
  31. Chen Y; Inaltekin H; Gorlatova M. Adapt SLAM: Edge-assisted adaptive SLAM with resource constraints via uncertainty minimization. IEEE INFOCOM 2023-IEEE Conference on Computer Communications. IEEE, 2023, 1-10.[CrossRef]
  32. Zhou C; Gao J; Li M, et al. Digital Twin-Based 3D Map Management for Edge-Assisted Device Pose Tracking in Mobile AR. IEEE Internet of Things Journal, 2024.[CrossRef]
  33. Wang K; Takemura N; Iwai D, et al. A typing assist system considering involuntary hand tremor. Transactions of the Virtual Reality Society of Japan, 2016, 21(2), 227-233.[CrossRef]
  34. Wang K; Iwai D; Sato K. Supporting trembling hand typing using optical see-through mixed reality. IEEE Access, 2017, 5, 10700-10708.[CrossRef]
  35. Ueda Y; Asai Y; Enomoto R, et al. Body cyberization by spatial augmented reality for reaching unreachable world. Proceedings of the 8th Augmented Human International Conference, 2017, 1-9.[CrossRef]
  36. Zhang X; Liu X; Yuan S M, et al. Eye Tracking Based Control System for Natural Human‐Computer Interaction. Computational intelligence and neuroscience, 2017, 2017(1),[CrossRef]
  37. Donaghy C; Thurtell M J; Pioro E P, et al. Eye movements in amyotrophic lateral sclerosis and its mimics: a review with illustrative cases. Journal of Neurology, Neurosurgery & Psychiatry, 2011, 82(1), 110-116.[CrossRef]
  38. De Gaudenzi E; Porta M. Towards effective eye pointing for gaze-enhanced human-computer interaction. 2013 Science and Information Conference, IEEE, 2013, 22-27.[CrossRef]
  39. Nehete M; Lokhande M; Ahire K. Design an eye tracking mouse. International Journal of Advanced Research in Computer and Communication Engineering, 2013, 2(2), 1118-1121.[CrossRef]
  40. Missimer E; Betke M. Blink and wink detection for mouse pointer control. Proceedings of the 3rd international conference on pervasive technologies related to assistive environments, 2010, 1-8.[CrossRef]
  41. Shahid E; Rehman M Z U; Sheraz M, et al. Eye Monitored Wheelchair Control. 2019 International Conference on Electrical, Communication, and Computer Engineering (ICECCE). IEEE, 2019, 1-6.[CrossRef]
  42. Xu C; Zhang X; Guan Q, et al. Analysis of the electrol physiological features of tremor and essential tremor in Parkinson's disease. Chinese Journal of Practical Nervous Diseases, 2017, 20(21),16-20.[CrossRef]

Round 2

Reviewer 2 Report

Comments and Suggestions for Authors

I have read the authors' replies to my review and the revised version of the manuscript. I am glad to see that the authors have by and large addressed my comments.

The only reply that I am not entirely satisfied with, is to my Comment 1, which relates to the choice of baselines for the comparison (no assistance at all). I do not quite agree that any tremor stabilization method can be so special that there's no way to compare it to the alternatives (particularly, considering the multitude of the performance parameters that the authors measure). However, I do not mind accepting the paper without the recommended additional baseline comparisons, provided that the authors clearly list the weakness of the employed baseline as a limitation. Currently, I do not see this in the Discussion section.

Another minor issue: I still see several "Error! Reference source not found" texts instead of references numbers - need to be fixed.

Author Response

Dear Reviewer,

Thank you for your continued engagement with our manuscript. We greatly appreciate the detailed and thoughtful feedback you have provided. Your comments have been instrumental in guiding our revisions and enhancing the overall quality of our work. We have diligently addressed each of your concerns and believe that the changes we have made have strengthened the manuscript. We value your expertise and are grateful for the opportunity to refine our research based on your insights.

Comments 1: [The only reply that I am not entirely satisfied with, is to my Comment 1, which relates to the choice of baselines for the comparison (no assistance at all). I do not quite agree that any tremor stabilization method can be so special that there's no way to compare it to the alternatives (particularly, considering the multitude of the performance parameters that the authors measure). However, I do not mind accepting the paper without the recommended additional baseline comparisons, provided that the authors clearly list the weakness of the employed baseline as a limitation. Currently, I do not see this in the Discussion section.]

Response 1: Thank you for your valuable feedback. We understand the importance of comparing our method with alternative tremor stabilization technologies. We have revised the Discussion section to clearly list the limitations of our employed baseline and acknowledge the absence of other tremor-assist technologies for comparison. Below is the revised text from the Discussion section:

[4.4. Discussion

The results of the first experiment demonstrate that the SAR system employing the virtual stabilization algorithm significantly reduced the time required for participants to complete remote target pointing tasks and markedly improved efficiency. Additionally, the average jitter distance of the virtual pointer during tasks was significantly reduced, further confirming the virtual stabilization algorithm’s effectiveness in enhancing the operational precision of the eye-controlled virtual hand and pointer. These findings underscore the critical role of the virtual stabilization algorithm in improving the stability and efficiency of interactions between individuals with tremor and distant objects through eye movement control.

In the second experiment, operations under the system-assisted condition significantly reduced errors compared to traditional manual interaction (Q4), highlighting the importance of the SAR system in enhancing operational precision, particularly when tremor patients use conventional control devices like TV remotes. However, there was no significant difference in ease of control (Q5) between the two conditions, suggesting that the system interface may not yet be fully optimized for eye movement interaction. Additionally, the results for operational stability (Q6) did not reach statistical significance, possibly due to insufficient sample size or the precision of the measurement tools.

The third experiment explored the performance of the SAR system in remote pointing tasks within complex environments. Supported by the system, participants achieved a median accuracy rate of 90%, and the median task completion time was only 3.65 seconds, demonstrating the system's ability to maintain high operational precision while ensuring rapid response. Survey results further indicated that the system significantly enhanced participants' ability to easily point to distant targets (Q7), improved the stability of pointing (Q8), clearly conveyed pointing intentions to peers (Q9), and substantially increased communication efficiency (Q10). These findings highlight the potential value of the SAR system in enabling precise and efficient interactions for tremor patients.

Despite these promising results, certain metrics, such as the ease of use of the SAR system in complex interaction tasks, did not show significant improvement. This suggests that future research should focus on developing assistive interface designs that are better suited to the characteristics of eye movement interactions to enhance user experience. Additionally, a significant limitation of this study is the absence of other tremor-assist technologies as comparative baselines. This omission prevents a comprehensive evaluation of the SAR system's relative advantages across various performance parameters. Therefore, future research should incorporate comparisons with other leading studies (such as [24] and [25]), including multiple baselines, to provide a more thorough assessment.

Although the complexity and diversity of head tremors are less than those of hand tremors, the virtual stabilization algorithm in this study effectively managed the head tremor frequencies and amplitudes of most common tremor patients. However, this design may not be sufficient for more severe types of tremors. Future systems should incorporate machine learning-based tremor measurement algorithms to assess mild, moderate, and severe tremors, thereby optimizing the selection and parameters of the virtual stabilization algorithm.

--These changes have been made on page [16]-page [17], line [627] to [669].]

Comments 2: [I still see several "Error! Reference source not found" texts instead of references numbers - need to be fixed.]

Response 2: We apologize for the oversight and appreciate your meticulous attention to detail. We have thoroughly reviewed the manuscript and corrected all instances of "Error! Reference source not found," ensuring that all references are now properly cited throughout the text.

We once again extend our gratitude to the reviewers for their comprehensive evaluation and constructive feedback. Your expertise and detailed comments have played a crucial role in improving the quality of our manuscript. We are confident that the revisions have addressed your concerns, and we thank you for your valuable input and consideration.
